

# Genome-wide identification, evolution, expression, and alternative splicing profiles of peroxiredoxin genes in cotton

Yulong Feng[1,*], Renhui Wei[2,*], Aiying Liu[2], Senmiao Fan[2], JinCan Che[1], Zhen Zhang[2], Baoming Tian[1], Youlu Yuan[2], Gongyao Shi[1] and Haihong Shang[1,2]

[1] Zhengzhou Research Base, State Key Laboratory of Cotton Biology, Zhengzhou University, Zhengzhou, China
[2] Chinese Academy of Agricultural Sciences, State Key Laboratory of Cotton Biology, Key Laboratory of Biological and Genetic Breeding of Cotton, The Ministry of Agriculture, Institute of Cotton Research, Anyang, China
[*] These authors contributed equally to this work.

## ABSTRACT

Peroxiredoxin (PRX) is a ubiquitous thioredoxin-dependent peroxidase that can eliminate excessive free radicals produced by stress and protect cells from oxidative damage. *PRX*s are also involved in reactive oxygen species (ROS)- and redox-dependent signaling by performing redox interactions with other proteins and modify their redox status. At present, *PRX* family identification, evolution and regulation research has been conducted in some plants; however, systematic research about this family is lacking in cotton. In this study, a total of 44 *PRX*s were identified in the cotton genome. Phylogenetic and conserved active site analyses showed that the *PRX*s were divided into six subfamilies according to the conserved site (PxxxTxxC…S…W/F) and conserved cysteinyl residues positions. Segmental duplication and polyploid events were the main methods for *PRX* family expansion, and the PRXs of diploid *G. arboreum* were the donors of *PRX*s in the D subgenomes of allotetraploid *G. hirsutum* and *G. barbadense* during the evolution of the *PRX* family. qRT-PCR analysis confirmed that cis-acting elements play important roles in regulating the expression of *PRX*s. Alternative splicing events occurred in GhPRX14-D that can increased the complexity of transcripts in *G. hirsutum*. Subcellular localization showed that most *PRX* members were located in chloroplasts, the cytoplasmic membrane and the nucleus. Our results provide systematic support for a better understanding of *PRX*s in cotton and a starting point for further studies of the specific functions of *PRX*s in cotton.

# INTRODUCTION

The peroxiredoxin (*PRX*) protein (EC 1.11.1.15) is a member of the thioredoxin-scaffold enzyme family and presents cysteine-dependent peroxidase activity against hydrogen peroxide substrates. *PRX* is ubiquitous among living organisms and has been found in animals, plants, protozoa, parasites, yeast, bacteria and archaea (*Knoops, Loumaye &*

Corresponding authors
Gongyao Shi, shigy@zzu.edu.cn
Haihong Shang, shanghaihong@caas.cn

*Eecken, 2007*). At present, research on this protein has mainly focused on mammalian disease resistance based on the loss of the protein, which leads to an imbalance in the cellular redox state (*Oláhová et al., 2009*). As an antioxidant enzyme, *PRX* is widely known to eliminate stress-induced excess free radicals in the body and protect cells from oxidative damage (*Hofmann, Hecht & Flohé, 2002*). *PRX* not only reduces the damage caused by oxidative stress but also enhances the activity of natural killer cells (*Nonn, Berggren & Powis, 2003*), regulates cell proliferation, differentiation and apoptosis (*Lee et al., 2003*), and protects free radical-sensitive proteins (*Wenders et al., 2003*).

In plants, a total of 10 *PRX* family members were previously identified in the Arabidopsis (*Arabidopsis thaliana*) genome and four were found to be targeted to chloroplasts (*Baier & Dietz, 1997*; *Horling, Baier & Dietz, 2001*), where they play important roles in chloroplast detoxification (*Dietz et al., 2006*). Some recent studies have demonstrated that 2-Cys*PRX* participates in chloroplast enzyme oxidation in the dark in Arabidopsis (*Pérez-Ruiz et al., 2017*). The Fd-FTR-Trxs and NTRC redox systems of chloroplasts are integrated via the redox balance of 2-Cys *PRX* (*Pérez-Ruiz et al., 2017*), and the chloroplast 2-Cys *PRX* functions as a thioredoxin oxidase in the redox regulation of chloroplast metabolism (*Pérez-Ruiz et al., 2017*). Subsequently, *Yoshida et al. (2018)* dissected the "dark side" of chloroplast redox regulation and provided insights into the thioredoxin-like2 (TrxL2)/2-Cys peroxiredoxin (2-CysPRX) redox cascade as a molecular basis for oxidative thiol modulation in chloroplasts. 2-CysPRX serves as an electron sink in the thiol network, which is important for the oxidation of reductively activated proteins and represents the missing link in the reversal of thioredoxin-dependent regulation (*Vaseghi et al., 2018*). All of these studies show that *PRX* plays important roles in the chloroplast redox system. In addition to its peroxidase function, 2-CysPRX has been proposed to be involved in the water-water cycle (WWC) and hydrogen peroxide ($H_2O_2$)-mediated signaling in plastids (*Awad et al., 2015*). The WWC is of particular importance in protecting the photosynthetic apparatus from photooxidative damage. Many proteins that interact with 2-CysPRX have been identified, such as thioredoxin-related electron donors (e.g., chloroplastic drought-induced stress protein of 32 kDa and atypical cysteine histidine-rich thioredoxin 2) and enzymes involved in chlorophyll synthesis (e.g., protochlorophyllide oxidoreductase B) or carbon metabolism (e.g., fructose-1,6-bisphosphatase) (*Cerveau et al., 2016*). In addition, *PRX* plays important roles in root growth (*Finkemeier et al., 2005*), photosynthesis protection (*Lamkemeyer et al., 2006*), antioxidant activity (*Pulido et al., 2010*) and drought tolerance (*Fichman et al., 2018*) in Arabidopsis. However, research on the *PRX* gene in cotton is lacking, and only a few studies have shown that the *PRX* gene is significantly upregulated in response to drought stress (*Zhang et al., 2016*). The availability of functional and structural information about *PRX* has increased rapidly in the past decade. For example, all the proteins in the family have conserved Cys residues at the N terminus and some members also have conserved Cys residues at the C terminus. These conserved Cys residues can be used to divide the *PRX* proteins into two subfamilies, namely, 1-CysPRX and 2-CysPRX (*Dietz, 2011*). However, the distinction between the functions of 2-Cys and 1-Cys is not particularly useful as a global classifier because representatives of each type seem to exist within all the subfamilies (*Hall et al., 2011*). Some studies have instead proposed classifying *PRXs*

based on structural and sequence information at the reactive cysteine active site (*Soito et al., 2011*), and the functional site profiling method (also referred to as conserved active site profiling) (*Nelson et al., 2011*) and the Deacon Active Site Profiler (DASP) tool (*Huff et al., 2005*) have been used to analyze the sequence conservation near the catalytic cysteine structure through bioinformatics analysis. Here, we compare the mechanistic method (using active structural sites) to the traditional method (determining the positions and numbers of conserved cysteinyl residues) to provide a global classification and localization of the *PRX* genes in cotton. In this study, we identified all of the *PRX* genes in four cotton species (*Gossypium hirsutum*, *Gossypium raimondii*, *Gossypium barbadense* and *Gossypium arboreum*), evaluated their evolutionary relationships and performed physical mapping to the chromosomes of each cotton species. In addition, we systematically analyzed the gene structures, conserved active sites and *cis*-acting elements of all the identified *PRX* genes in these four cotton species. To explore the functions of abiotic stress and hormone-induced *cis*-acting elements in the *PRX* gene promoters, we carried out experiments on the expression trends of the *PRXs* under different stresses. The expression levels of *PRXs* in cotton tissues and organs were analyzed using *G. hirsutum* TM-1 transcriptome data (*Zhang et al., 2015*), and the alternative splicing (AS) profiles of the *PRXs* were identified and verified by RT-PCR. Finally, to determine the localizations of the *PRX* proteins in cells, we carried out subcellular localization experiments with *Agrobacterium tumefaciens*-infected tobacco. These results provide a solid foundation for the study of the distribution, structure and evolution of the cotton *PRX* family, and the regulatory mechanisms, transcript abundances and cellular localizations of these *PRXs* will provide important information for follow-up studies of their functional differentiation and applications.

## MATERIALS AND METHODS

### Sequence sources

*G. raimondii* (*Gossypium raimondii* L.,JGI) (Accession: PRJNA171262) (*Paterson et al., 2012*) and *G. hirsutum* (*Gossypium hirsutum* L., NAU) (Accession: PRJNA248163) (*Zhang et al., 2015*) genomic data files were obtained from the JGI database (https://genome.jgi.doe.gov/portal/); *G. arboreum* (*Gossypium arboreum* L., CRI) (Accession: PRJNA382310) (*Du et al., 2018*) and *G. barbadense* (*Gossypium barbadense* L., Hau) (Accession: PRJNA433615) (*Wang et al., 2018*) genomic data files were downloaded from the CottonGen database (https://www.cottongen.org/).

The following protein sequence data were obtained from the corresponding databases. Arabidopsis (*Arabidopsis thaliana* L.) (Accession: SRA009031) (*Filichkin et al., 2010*) protein sequence data were obtained from The Arabidopsis Information Resource (http://www.arabidopsis.org). Rice (*Oryza sativa* L.) (*Ouyang et al., 2007*) protein sequence data were downloaded from the Rice Genome Annotation Project (http://rice.plantbiology.msu.edu/index.shtml). Cacao (*Theobroma cacao* L.) (Accession: PRJNA51633) (*Motamayor et al., 2013*) protein sequence data were obtained from the JGI database (https://genome.jgi.doe.gov/portal/), and grapevine (*Vitis vinifera* L.) (GenBank:

CU459218–CU462737) (*Jaillon et al., 2007*) protein sequence data were obtained from the Ensembl Plants database (http://plants.ensembl.org/index.html).

## Identification and conserved active site analysis of the *PRX* family in cotton

The *PRX* Pfam domain ids (http://pfam.xfam.org/family/PF08534.9, http://pfam.xfam.org/family/PF00578.20, http://pfam.xfam.org/family/PF10417.8) in cotton were identified using the *PRX* protein sequence of *G. hirsutum* on the EMBL-EBI (https://www.ebi.ac.uk/Tools/hmmer/) website. The Pfam model files were downloaded from the Pfam database (http://pfam.xfam.org/) (*Finn et al., 2009*). The protein databases of four cotton species (*G. raimondii*, *G. arboreum*, *G. hirsutum*, and *G. barbadense*) and Arabidopsis, Rice, Cacao, and Grapevine were searched with hmmsearch (v3.2.1) (https://www.ebi.ac.uk/Tools/hmmer/search/hmmsearch) ($E <= 0.001$). All possible *PRX* genes were identified in these eight crop species. The BLAST program was used to compare the sequences to the Arabidopsis protein database ($p <= 1E^{-10}$), and then, the false positive *PRX* genes were deleted on the basis of the Arabidopsis protein annotation file (https://www.arabidopsis.org/). To further ensure the accuracy of each candidate *PRX*, ClustalW in the MEGA7 (https://www.megasoftware.net/) software was used to compare the candidate *PRX* protein sequences with the conserved active sites and positions of conserved cysteinyl residues in the *PRX* proteins in the *PRX* protein database (http://csb.wfu.edu/prex/) (*Dietz, 2011*; *Nelson et al., 2011*) and delete sequences with no conserved active site (PxxxTxxC…S…W/F). Blastp alignment of the identified *PRX* proteins from the NCBI reference protein database was performed (percent identity $\geq$90%). The theoretical isoelectric point (pI) and molecular weight (MW) of the *PRX* proteins were investigated with ExPASy (http://web.expasy.org/protparam/) (*Finn et al., 2014*).

## Phylogenetic and gene structure analysis

ClustalW in MEGA 7.0 software (*Kumar, Stecher & Tamura, 2016*) was used to compare the protein sequences, and a phylogenetic analysis was then performed. A phylogenetic tree was constructed in MEGA7 using the maximum likelihood (ML) method, the Poisson correction model, complete deletion and bootstrap analysis performed with 1,000 replicates.

The exon/intron structures of the *PRXs* were extracted from the corresponding cotton GFF files by TBtools software (http://www.omicshare.com/forum/thread-1062-1-1.html), and the *PRX* structure maps were drawn by GSDS 2.0 (http://gsds.cbi.pku.edu.cn/) (*Hu et al., 2014*). MEME (4.11.4) (http://meme-suite.org/) (*Bailey et al., 2006*) was used to identify the conserved motifs of the cotton *PRXs* with the following parameters: the maximum number of motifs was 10 and the optimal width was 6 $\leq$250. Then, TBtools was used to construct a vector graph from the xml file generated by MEME.

## Chromosomal mapping and gene duplication

To reveal the chromosomal mapping and duplication relationships of *PRXs* in cotton, genome databases for *G. raimondii*, *G. arboreum*, *G. hirsutum* and *G. barbadense* were constructed. The physical positions of *PRXs* in cotton were fetched from the corresponding

GFF files. *PRX* family duplication data were identified by the MCScanX program ($p \leq 1E^{-20}$) (*Ding et al., 2015*), and a comparative genome analysis of the *PRX* family genes in these four cotton species was carried out ($p \leq 1E^{-20}$) (*Wang, Li & Paterson, 2013*). Visualization was carried out with the CIRCOS (http://circos.ca/) tool (*Krzywinski et al., 2009*). The substitution rates of synonymous (Ks) and nonsynonymous (Ka) sites were calculated by the KaKs calculator program (*Suyama, Torrents & Bork, 2006*). The divergence time was calculated by the formula T=Ks/2$^{\lambda}$ ($\lambda = 1.5 \times 10^{-8}$) (*Zhang et al., 2006*), where Ks was the synonymous substitution of each locus and r was the divergence rate of plant genes. For dicotyledonous plants, r is considered to be $1.5 \times 10^{-8}$ synonymous substitutions per site per year (*Koch, Haubold & Mitchell-Olds, 2000*).

## Analysis of *cis*-acting elements in the promoter region

By using the genome data files (GFF3) of the four cotton species, the promoter sequences of the *PRX* genes (2,500 bp upstream of the initiation codon "ATG") were extracted from the cotton genome sequences (*Wang et al., 2012*). PlantCARE (http://bioinformatics.psb.ugent.be/webtools/plantcare/html/search_CARE.html) (*Lescot et al., 2002*) was used to predict the *cis*-acting elements of the promoter sequences, and the abiotic stress response, plant growth and development, and then the hormone-induced *cis*-acting elements were analyzed (Table S2).

Based on the *cis*-acting elements identified in the promoter sequence, we used GSDS 2.0 (http://gsds.gao-lab.org/) (*Hu et al., 2014*) to display the physical sites of the *cis*-acting elements in the *G. hirsutum* promoter sequence.

## Plant material treatments and expression analysis

To analyze the expression patterns of the *PRX* genes, PEG, NaCl and salicylic acid (SA) were used to simulate drought stress, osmotic stress and SA hormone induction, respectively. The seeds of *G. hirsutum* sGK9708 were soaked in flasks overnight, germinated in fine sand at 28 °C in the dark for 2 days, and transferred to a greenhouse for hydroponic growth. The hydroponic greenhouse conditions were as follows: 28 °C day/25 °C night, 14-hour photoperiod and 70% relative humidity. Plants were grown in Hoagland nutrient solution (*Xing et al., 2019*). At the three-leaf stage, the seedlings were subjected to stress treatments. The cotton seedlings were equally divided into four groups and transferred to nutrient solutions of the hydroponic box supplemented with 200 mM sodium chloride (NaCl), 15% PEG-6000, 0.1 mM SA and a blank control for the salt, drought, hormone induction and control treatments, respectively. A total of 20 cotton seedlings received each treatment, and each treatment was repeated three times. At 0, 1, 3, 6, and 12 h of treatment, four seedlings were randomly selected from four treatment groups at each time point, and the leaves (second true leaf) and rhizomes were removed from the seedlings and immediately frozen in liquid nitrogen. The leaves were stored at −80 °C before RNA extraction.

Total RNA was isolated using an EasyPure Plant RNA Kit (TransGen, Beijing, China), and RNA reverse transcription was performed using a TransScript reverse transcription system (AT341). *Gh-PRX*-specific primers (Table S3) were used to find candidate-specific primers on the qPrimerDB-qPCR Primer Database (https://www.ncbi.nlm.nih.gov/) website. To

ensure the specificity of the *Gh-PRX* gene primers, the candidate-specific primers were subjected to BLAST homologous comparison in the Primer-BLAST database of NCBI (https://www.ncbi.nlm.nih.gov/) (National Center for Biotechnology Information) to ensure that the specific primers amplified only their respective target gene fragments (Table S3). The general fluorescent dye mixture used was SYBR Green I (TransStart), and the reactions were performed with a 7500 Rapid Real-time PCR system (Roche). Ubiquitin 7 (UBQ7) (GenBank: DQ116441) was used as the internal standard reference to measure the expression levels of the cDNA genes. The volume of each reaction was 10 µl, and the reaction conditions were as follows: 94 °C for 5 min and 40 cycles of 94 °C for 5 s, 55 °C for 30 s and 72 °C for 30 s. During the extension step, the fluorescence signal was measured, and the acquisition time was set to 30 s. Each cDNA sample was repeated three times, and the results were analyzed by the $2^{-\Delta CT}$ method (*Khan-Malek & Wang, 2011*).

## Analysis of tissue expression and AS profiles of *PRX*s

We used published *G. hirsutum* (TM-1) (*Zhang et al., 2015*) transcriptome data, including data from the roots, stems, leaves, cotyledons, petals, stamens, ovules, seeds, and 5, 10, 15 and 20 DPA fibers. The expression levels of *Gh-PRX* genes in various tissues were calculated using log2 (FPKM) values (*Zhang et al., 2015*). The expression values were normalized by Genesis software and illustrated with a heatmap by HemI 1.0 - Heatmap illustrator software (*Sturn, Quackenbush & Trajanoski, 2002*).

Samples of *G. hirsutum* sGK9708 RNA without treatment at −80 °C were obtained. Total RNA was isolated using the EasyPure Plant RNA Kit (TransGen, Beijing, China), and RNA reverse transcription was performed using the TransScript reverse transcription system (AT321). The obtained first-strand cDNA was used for subsequent RT-PCR amplification. The primers for synthesizing full-length cDNA were designed using Primer Premier 5.0 software (Premier Biosoft International, Palo Alto, CA). After RT-PCR, the amplification products were detected by 1.2% agarose gel electrophoresis. After recovery and purification, the PCR products were ligated with the pEASY-Blunt cloning vector (TransGen, Beijing, China) (The pEASY®-Blunt cloning kit was purchased from TransGen Biotech, catalog number: CB101-01) (https://www.transgenbiotech.com/) and transformed into *E. coli* (*Escherichia coli*)-competent DH5 α cells (TransGen, Beijing, China). Thirty individual colonies were selected for positive PCR identification. The positive colonies were sequenced (Sangon, Shanghai), and the full-length sequences were obtained. These sequences were compared by Splign (https://www.ncbi.nlm.nih.gov/sutils/splign/splign.cgi) and GSDS 2.0 (http://gsds.gao-lab.org/), and the AS events were displayed on the gene structure.

## Subcellular localization of *PRX* proteins

The TargetP 1.1 (http://www.cbs.dtu.dk/services/TargetP/) server website (*Emanuelsson et al., 2007*) was used to predict and analyze the localization of *G. hirsutum PRX* proteins in plant cells.

To verify the distribution of *PRXs* in plant cells, five pairs of homologous genes from among 15 *Gh-PRX* genes were selected, and the full-length coding sequences (CDSs) of the *PRX* genes with no terminator codon and restriction site were amplified and

subcloned into the pCAMBIA2300-35S-eGFP (CAMBIA) transient expression vector (The pCAMBIA2300-35S-eGFP vector was assembled by inserting the 35S-eGFP fragment into the pCAMBIA2300 vector skeleton in our laboratory). The accuracy of the 10 recombinant vectors was ensured by primer-based PCR and sequencing analysis. Then, we cultured *Nicotiana benthamiana* seedlings at 28 °C day/25 °C night with a 14-hour photoperiod and 70% relative humidity for 1 month. When the third or fourth leaf was dark green and sufficiently thick, *Agrobacterium tumefaciens* harboring the recombinant vector was injected into the tobacco leaves (*Poulsen, Palmgren & López-Marqués, 2016*) to introduce the recombinant vector into tobacco leaf epidermal cells, and the plants were cultured in darkness at 25 °C for 24 h and then under light for 24 h. A square slice of 1 cm$^2$ was cut next to the injection area, placed on a slide with PBS buffer, and then gently covered with a cover glass. The position of the fluorescent protein in tobacco epidermal cells was observed by laser confocal microscopy (Olympus FV1200). The pCAMBIA2300-eGFP (CAMBIA) empty vector without the *Gh-PRX* gene was used as a control. The nuclei were stained with DAPI solution (Solarbio) as a control (*Kapuscinski, 1995*).

## RESULTS

### Genome-wide identification and conserved motif analysis of the *PRX* family in cotton

To identify the *PRX* genes of cotton, we used the redoxin domain, 1-cys-PRX_C domain and AhpC-TSA domain as references. The protein databases of four cotton species were searched by Hmmsearch (*Bhaduri, Ravishankar & Sowdhamini, 2004*). Based on the conserved active sites and the positions of conserved cysteinyl residues, 44 *PRX* genes were retrieved (Table 1), of which 15, 8, 8 and 13 *PRX*s were identified from *G. hirsutum, G. raimondii, G. arboreum* and *G. barbadense*, respectively. The PRXs were then divided into six subfamilies, named 2-CysPRX, 1-CysPRX, PRQ, PRXIIB, PRXIIE and PRXIIF, according to their conserved active sites, the positions of their conserved cysteinyl residues and the relevant literature (*Dietz, 2011*; *Nelson et al., 2011*) (Fig. 1).

We performed multiple sequence alignment of the 44 *PRX* proteins using ClustalW in MEGA7 and identified in detail the specific and conserved active sites of the *PRX* proteins (*Adak & Begley, 2017*). The results showed that the active sites of the *PRX* protein sequences were consistent with those (Fig. 1) by which the *PRX* family was classified in the PREX database (http://csb.wfu.edu/prex/). All PRX family proteins have conserved P, T, C, S and W/F amino acid sites, and the fifth conserved amino acid in the PRXQ subfamily protein sequence is F (Phe). In addition, we refer to the position and number of conserved cysteine residues in the classification of subfamilies (*Dietz, 2011*). 2-CysPRX has 2 Cys residues: Cys$_P$ (FFYPLDFTFVCPTEI) is generally located in the N-terminal part of the protein at approximately amino acid (aa) position 118, and Cys$_R$ (EVCP) is located in the C-terminal part of the protein at approximately aa position 240 (Fig. 1). The two catalytic Cys residues in PRXQ are separated by 4 amino acids (Fig. 1) (*Kong et al., 2000*). The two catalytic residues of PRXII, Cys$_P$ and Cys$_R$, are separated by 24 amino acids. The subclassification of PRXII proteins has always been based on their subcellular localization

**Table 1** The characteristics of the *PRX* family genes in cotton.

| Gene name | Gene ID | Subcalade | Exon | Protein (aa) | pI | Mw(Dalton) | Chromosome location |
|---|---|---|---|---|---|---|---|
| GhPRX1-A | Gh_A01G1793 | PRX2 | 1 | 228 | 9.1 | 24095.74 | ChrA01:97705624-97706307 |
| GhPRX2-A | Gh_A04G0493 | PRX1 | 7 | 266 | 6.19 | 29149.42 | ChrA04:26311787-26315757 |
| GhPRX3-A | Gh_A05G3849 | TPX | 3 | 163 | 5.6 | 17466.1 | ChrA05:104118-105049 |
| GhPRX4-A | Gh_A08G2151 | TPX | 3 | 163 | 5.58 | 17226.79 | ChrA08:102615361-102616794 |
| GhPRX5-A | Gh_A08G2202 | PRXQ | 4 | 217 | 9.74 | 23887.43 | ChrA08:103031734-103033703 |
| GhPRX6-A | Gh_A10G1104 | PRX5 | 5 | 199 | 8.98 | 21446.44 | ChrA10:54462313-54464296 |
| GhPRX7-A | Gh_A10G1567 | PRX6 | 4 | 220 | 5.98 | 24555.88 | ChrA10:85538436-85539624 |
| GhPRX8-D | Gh_D01G2034 | PRX2 | 1 | 228 | 9.1 | 24111.83 | ChrD01:59334370-59335053 |
| GhPRX9-D | Gh_D04G0917 | PRX1 | 7 | 266 | 6.19 | 29043.3 | ChrD04:26446572-26450644 |
| GhPRX10-D | Gh_D05G0244 | TPX | 3 | 147 | 6.82 | 15755.16 | ChrD05:2202510-2204565 |
| GhPRX11-D | Gh_D05G1251 | TPX | 3 | 163 | 5.78 | 17454.09 | ChrD05:10748631-10749537 |
| GhPRX12-D | Gh_D08G2518 | TPX | 3 | 163 | 5.58 | 17196.76 | ChrD08:64930986-64932473 |
| GhPRX13-D | Gh_D08G2567 | PRXQ | 4 | 217 | 9.7 | 23793.27 | ChrD08:65350134-65351542 |
| GhPRX14-D | Gh_D10G1403 | PRX5 | 5 | 199 | 8.45 | 21361.29 | ChrD10:28600472-28602454 |
| GhPRX15-D | Gh_D10G1825 | PRX6 | 4 | 220 | 6.22 | 24392.79 | ChrD10:51342185-51343360 |
| GbPRX1-A | Gbar_A10G018830.1 | PRX6 | 4 | 219 | 5.98 | 24555.88 | ChrA10:93866990-93868166 |
| GbPRX2-A | Gbar_A08G026850.1 | PRXQ | 4 | 216 | 9.74 | 23887.43 | ChrA08:119243921-119246359 |
| GbPRX3-A | Gbar_A08G026350.1 | TPX | 3 | 162 | 5.58 | 17196.76 | ChrA08:118790855-118792723 |
| GbPRX4-A | Gbar_A01G020700.1 | PRX2 | 1 | 227 | 9.1 | 24095.74 | ChrA01: 113395150-113395833 |
| GbPRX5-A | Gbar_A10G013540.1 | PRX5 | 5 | 198 | 8.98 | 21446.44 | ChrA10:62896091-62898528 |
| GbPRX6-A | Gbar_A05G012820.1 | TPX | 3 | 162 | 5.6 | 17466.1 | ChrA05:11767776-11768707 |
| GbPRX7-D | Gbar_D10G014760.1 | PRX5 | 5 | 198 | 8.45 | 21333.23 | ChrD10:28146247-28148956 |
| GbPRX8-D | Gbar_D04G010400.1 | PRX1 | 7 | 265 | 6.19 | 29103.4 | ChrD04:28395391-28400177 |
| GbPRX9-D | Gbar_D08G027530.1 | PRXQ | 4 | 216 | 9.7 | 23793.27 | ChrD08:65405467-65407385 |
| GbPRX10-D | Gbar_D01G021840.1 | PRX2 | 1 | 227 | 9.34 | 24091.84 | ChrD01:60905018-60905701 |
| GbPRX11-D | Gbar_D08G027000.1 | TPX | 3 | 162 | 5.58 | 17196.76 | ChrD08:65009751-65011737 |
| GbPRX13-D | Gbar_D05G002520.1 | TPX | 3 | 146 | 6.82 | 15755.16 | ChrD05:2189964-2198116 |
| GbPRX14-D | Gbar_D10G019020.1 | PRX6 | 4 | 219 | 6.22 | 24392.79 | ChrD10:51564063-51565371 |
| GrPRX1 | Gorai.009G135600.1 | TPX | 3 | 162 | 5.78 | 17468.12 | Chr9:10193477-10194845 |
| GrPRX2 | Gorai.009G397800.1 | PRX1 | 7 | 265 | 6.19 | 29043.3 | Chr9:56100514-56105218 |
| GrPRX3 | Gorai.009G026300.1 | TPX | 3 | 163 | 5.9 | 17491.13 | Chr9:2005154-2007919 |
| GrPRX4 | Gorai.011G158200.1 | PRX5 | 10 | 198 | 7.71 | 21319.16 | Chr11:28114223-28116745 |
| GrPRX5 | Gorai.011G204600.1 | PRX6 | 4 | 219 | 5.98 | 24498.87 | Chr11:49579270-49580743 |
| GrPRX6 | Gorai.002G243200.1 | PRX2 | 1 | 227 | 9.24 | 24170.9 | Chr2:60733892-60734923 |
| GrPRX7 | Gorai.004G280800.1 | TPX | 3 | 162 | 5.58 | 17196.76 | Chr4:61267661-61269611 |
| GrPRX8 | Gorai.004G285400.1 | PRXQ | 4 | 216 | 9.7 | 23793.27 | Chr4:61634457-61636274 |
| GaPRX1 | Ga05G1380 | TPX | 3 | 162 | 5.78 | 17500.12 | Chr05:12175377-12176307 |
| GaPRX2 | Ga10G1482 | PRX5 | 5 | 198 | 8.98 | 21445.49 | Chr10:80207013-80208997 |
| GaPRX3 | Ga05G0262 | TPX | 3 | 163 | 5.58 | 17538.2 | Chr05:2267143-2271319 |
| GaPRX4 | Ga04G0982 | PRX1 | 7 | 265 | 6.19 | 29045.27 | Chr04:36972413-36976371 |
| GaPRX5 | Ga02G1606 | PRX2 | 1 | 227 | 9.1 | 24135.81 | Chr02:97187166-97187849 |
| GaPRX6 | Ga08G2895 | PRXQ | 4 | 216 | 9.7 | 23846.33 | Chr08:128777936-128779888 |
| GaPRX7 | Ga08G2842 | TPX | 3 | 162 | 5.58 | 17196.76 | Chr08:128370617-128372047 |
| GaPRX8 | Ga10G0941 | PRX6 | 5 | 219 | 5.96 | 22794.86 | Chr10:20870875-20872058 |

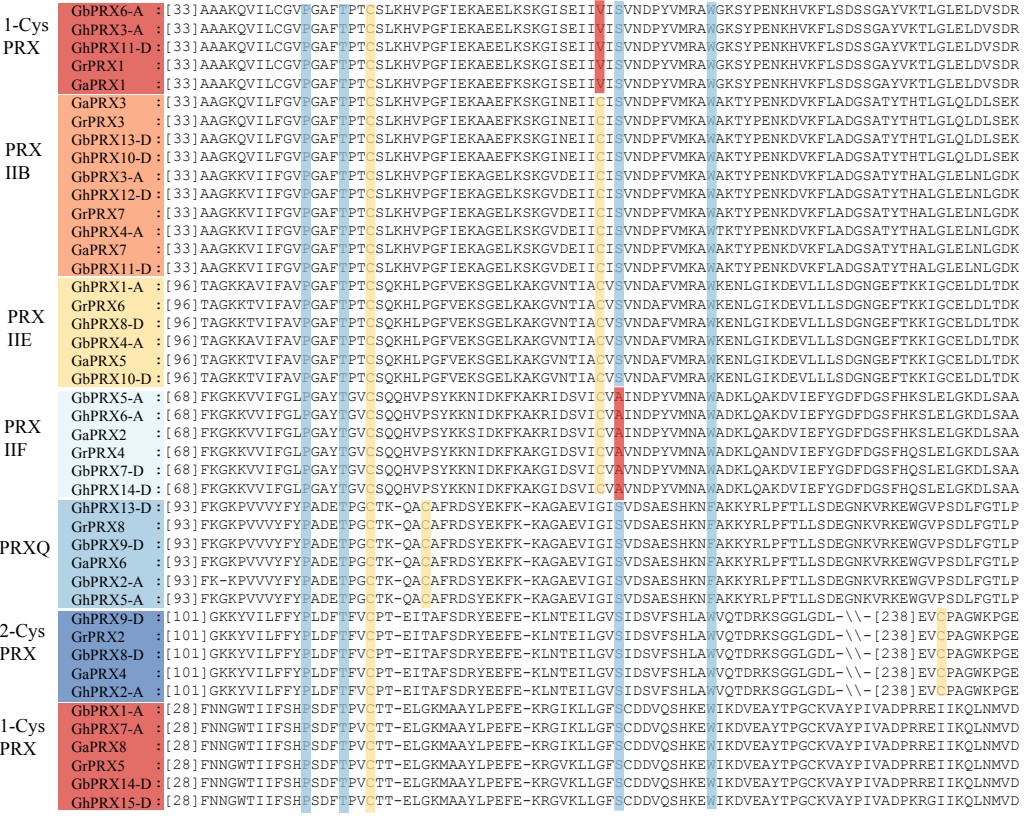

**Figure 1** **The conserved active sites of the *PRX* protein sequence in cotton.** The multiple sequence alignment of 44 *PRX* proteins in cotton using ClustalW in MEGA7. The active conserved sites and positions of conserved cysteinyl residues were marked with columnar bars. The conserved cysteinyl residues marked by yellow bars. The mutation site is marked by red bars.

(*Dietz, 2011*), In addition, we compared the known PRXII proteins in Arabidopsis, rice and cotton and analyzed the differences between their conserved sequences, which facilitates a more accurate classification of PRXs (Fig. S1). PRXIIE proteins are localized in the chloroplast (*Mhamdi & Van Breusegem 2018*; *Romero-Puertas et al., 2007*) and share the conserved sequences (FGLPGAYTGVCSQ....C.S...W). PRXIIF proteins are localized to the mitochondria (*Horling, König & Dietz, 2002*), and display a conserved motif (Fig. S1). Compared with the PRXIIF protein sequence in Arabidopsis, the S (serine) conserved was mutated to A (alanine) in cotton. The second cysteinyl (Cys) in the protein sequences of some genes originally predicted to belong to the PRXIIB subgroup (GbPRX6-A, GhPRX3-A, GhPRX11-D, GrPRX1 and GaPRX1) was mutated to V (Valine) (Fig. 1) (Table S5), resulting in these genes having a cysteine residue only at N-terminal amino acid position 50 and being classified as 1-CysPRX. Typical 1-CysPRX proteins do not have a second Cys (Fig. 1) (*Dietz, 2011*) and only have CysP followed by a peroxide domain.

The MW and pI of the PRXs were calculated using the Compute pI/Mw tool on the ExPASy (https://web.expasy.org/compute_pi/) website. The encoded protein lengths of the 44 PRX genes varied from 100 to 300 amino acids, the predicted MWs were 13-30

kilodaltons (kDa), and the pI values were approximately 4.0–10.0. Subcellular localization analysis predicted that 17 of the 44 proteins localized to the chloroplasts (Table S4); thus, these members of the PRX family may be involved in regulating the redox status of chloroplast proteins.

## Phylogenetic and gene structure analysis of *PRX*s in cotton

We constructed a phylogenetic tree using 74 *PRX*s from eight dicot species (G. raimondii, G. arboreum, G. hirsutum, G. barbadense, Arabidopsis, rice, cacao, grapevine). The results show that the cotton PRX genes are divided into six subfamilies: 2-CysPRX, PRXQ, PRXIIE, PRXIIB, PRXIIF and 1-CysPRX (Fig. 2). Moreover, the cotton *PRX* genes were clustered within every clade, indicating that these genes had undergone specific expansions in all six subfamilies. A conserved domain analysis helped to clarify the functional evolution and evolutionary relationships of cotton *PRX*s. A total of 10 conserved motifs were identified in the *PRX* proteins of the 4 cotton species. The motif logo of these conserved motifs is shown in Fig. S2, and the flags for the 10 conserved motifs are shown in Fig. 3C. The number of conserved motifs in each PRX protein ranged from 4 to 9. There were four motif fragments containing conserved active structural sites (motif 1, motif 4, motif 5 and motif 6) (Fig. 3C), and the details can be seen in Fig. S3. The conserved structural sites in the PRXIIB, PRXIIE, PRXIIF, 1-CysPRX and PRXQ subfamilies are located in motif 1, while the conserved structural sites in 2-CysPRX and 1-CysPRX are composed of motif 4, motif 5 and motif 6. Compared with the motifs containing conserved active sites, the seven other motifs (2, 3, 7, 8, 9, 10) play important roles in the functional differentiation of *PRX* family genes.

Exon-intron structural differences play a key role in the evolution of polygene families. We found that the number of introns in the 44 *PRX* genes ranged from 0 to 6. *PRX*s in the same subfamily showed a similar exon-intron structure (Fig. 3B). The PRXIIB subfamily contained five intron sequences of different lengths. PRXIIE has no introns, and PRXIIF has 4 introns. PRXQ contained three introns, and most of the 2-CysPRX and 1-CysPRX subfamily genes also contained the same number of introns. However, there was an intron sequence before the translation initiation site (TIS) in GbPRX5-A and GbPRX14-D. Introns before the TIS are integral parts of genes and play important roles in regulating gene expression (*Li, Li & Zhang, 2013*).

## Chromosome mapping and duplication analysis

The *PRX* genes were physically mapped to the chromosomes of *G. hirsutum* and *G. barbadense*. We found that the mapping of *PRX*s to the chromosomes of these two cotton species was similar (Figs. 4A and 4B), which shows that the genes of this family were conserved during evolution. However, the homologs of three of these PRX genes (GbPRX6-A, GhPRX11-D and GbPRX8-D), which are marked in black in the figure, may have been lost during evolution. There are six pairs of homologous genes in the G. hirsutum PRX family, which are distributed on chromosomes Gh_A01, Gh_A04, Gh_A08, Gh_A10, Gh_D01, Gh_D04, Gh_D08 and Gh_D10. Among the 15 Gh-PRX genes, GhPRX3-A was localized to scaffolds, and its exact position has not yet been determined. There are five
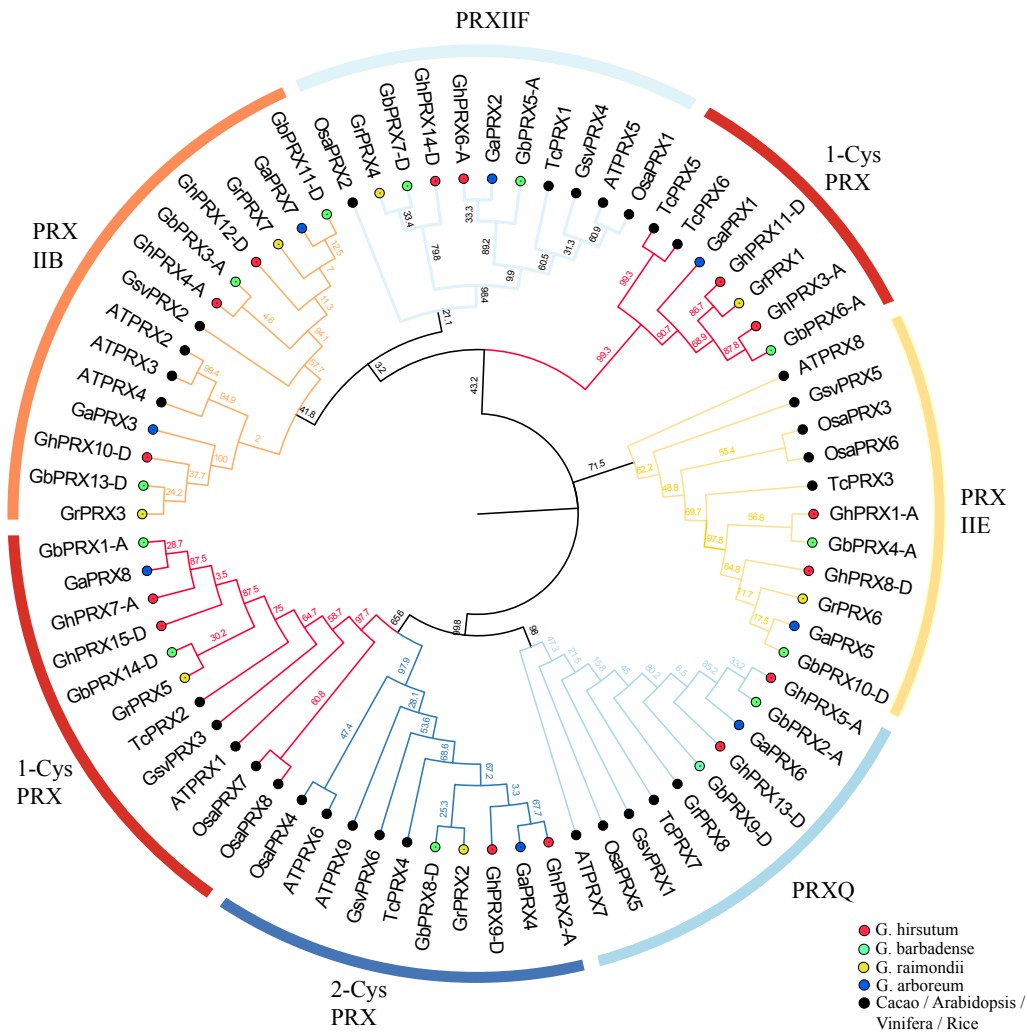

**Figure 2 Phylogenetic analysis of *PRX* family members.** A maximum likelihood phylogenetic tree was constructed with *PRX* protein sequences from *G. hirsutum*, *G. barbadense*, *G. arboreum*, *G. raimondii*, *Arabidopsis*, *Rice*, *Cacao*, and Grapevine. Using the Poisson correction model, complete deletion and bootstrap analysis with 1,000 replicates.

pairs of PRX homologous genes in G. barbadense, which were mapped to chromosomes Gb_A01, Gb_A08, Gb_A10, Gb_D01, Gb_D08 and Gb_D10. Compared with those of G. hirsutum, the PRXs on Gb_A04 of G. barbadense were lost; therefore, one pair of homologous genes was missing. The comparative genomic analysis of PRXs between G. hirsutum and G. barbadense showed that GbPRX6-A on Gb_A05 originated from a translocation of GhPRX11-D on Gh_D05 (Fig. 4B and Fig. S4). Based on the physical chromosome mapping analysis and comparison of PRXs between the G. arboreum and G. raimondii genomes, we found that only the location of GaPRX5 and GrPRX6 on Chr02 was relatively conserved in these two cotton species (Figs. 4C–4D and Fig. S4).

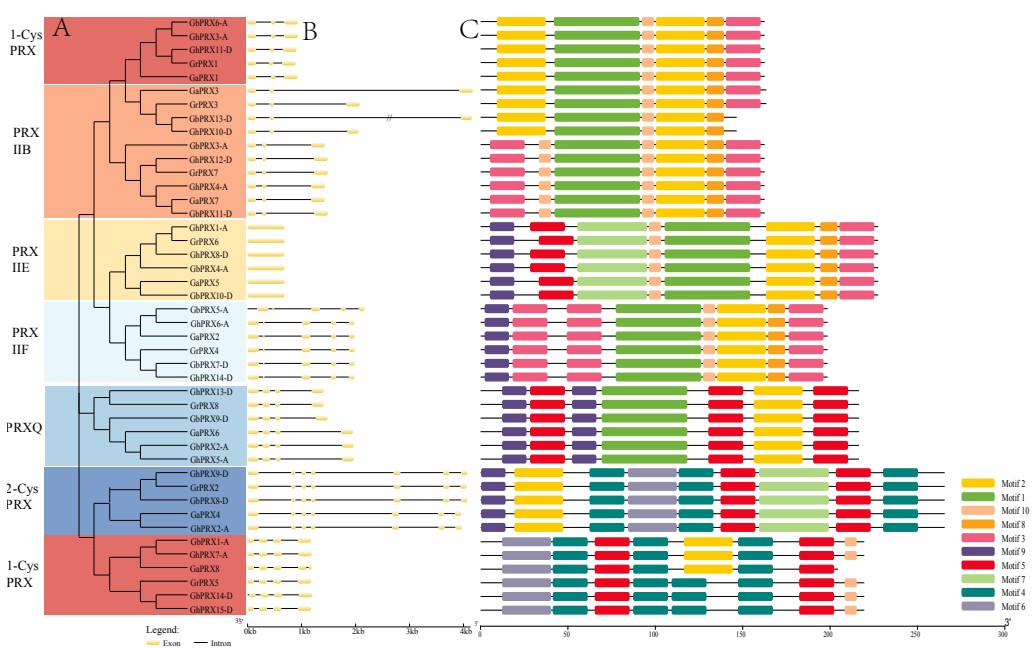

**Figure 3** **Phylogenetic tree, gene structure and conserved motif of *PRXs* in cotton.** (A) Phylogenetic tree was constructed in MEGA7 using maximum likelihood (ML) method, the Poisson correction model, complete deletion and bootstrap analysis performed with 1,000 replicates. (B) The exon/intron distribution of *PRX* genes. (C) The distribution of domains in *PRX* proteins.

We identified eight pairs of duplicates among the 15 PRXs of G. hirsutum (GhPRX8-D:GhPRX1-A, GhPRX9-D:GhPRX2-A, GhPRX10-D:GhPRX12-D, GhPRX10-D:GhPRX4-A, GhPRX12-D:GhPRX4-A, GhPRX5-A:GhPRX13-D, GhPRX15-D:GhPRX7-A, and GhPRX14-D:GhPRX6-A) (Fig. 4A), and these duplicates are located on six pairs of chromosomes (Gh_A01:Gh_D01, Gh_A04:Gh_D04, Gh_D05:Gh_D08, Gh_A08:Gh_D05, Gh_A08:Gh_D08, and Gh_A10:Gh_D10). With the exception of GhPRX10-D:GhPRX12-A and GhPRX10-D:GhPRX4-A, which underwent segmental duplication, these duplicates were produced during the process of genomic polyploidization. Compared with those in G. hirsutum, the duplicated genes on the *G. barbadense* Gb_A04 and Gb_D04 chromosomes were altered (Fig. 4B). GbPRX8-D on the Gb_D04 chromosome was conserved, although the corresponding homologous gene on Gb_A04 was lost during evolution. The other duplicated genes were the same as those in *G. hirsutum* and remained conserved. The *PRX*s of the diploid *G. arboreum* genome are very similar to the PRXs of the tetraploid *G. hirsutum* and *G. barbadense* D subgenome, only GaPRX5 was lost during evolution, and segmental duplication genes on Chr05 and Chr08 chromosomes were highly homologous (Fig. 4D and Fig. S4). However, in *G. raimondii*, these segmentally duplicated genes were physically located on the Gr_chr04 and Gr_chr09 chromosomes (Fig. 4C). According to a KaKs calculation of the duplicated gene pairs in *G. raimondii* and *G. arboreum* (Table S1), the divergence time of the duplicated genes (GrPRX3:GrPRX7) on the *G. raimondii* chromosomes Gr_chr04 and Gr_chr09 was 24.81 MYA, while the divergence time of the

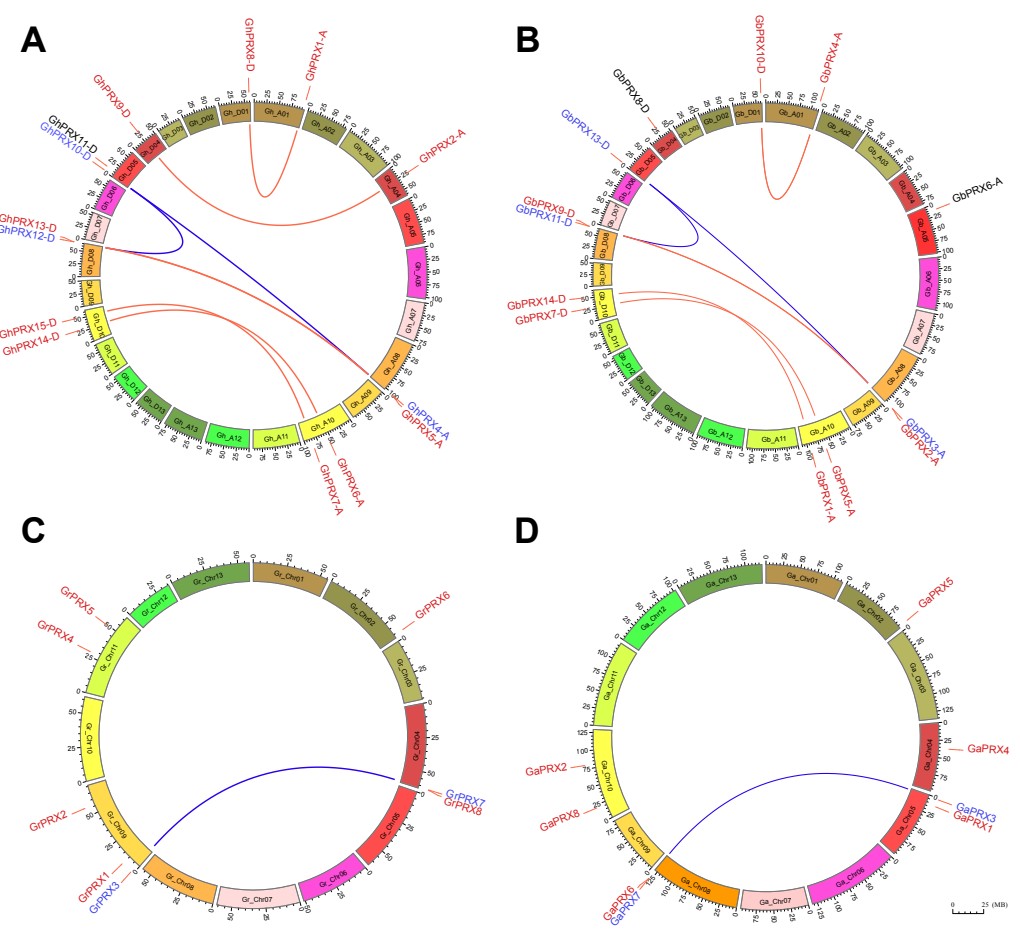

**Figure 4 Chromosome physical mapping and duplication events of *PRX* genes in the genomes of four cotton species.** Different color blocks represent different chromosomes, and the minimum scale represents 1,000,000 chromosome units. red links mark the physical site of genes on chromosomes. The homologous genes of the genes marked in black have been lost during the evolution of the species. The segmental duplication event occurred in the genes marked in blue. The homologous genes produced by segmental duplication events are connected by a blue curve, and the homologous genes produced by genomic polyploidization events are connected by a red curve. (A) Chromosomal physical mapping and duplication events of *PRXs* in the *G. hirsutum* genome. (B) Chromosomal physical mapping and duplication events of *PRXs* in the *G. barbadense* genome. (C) Chromosomal physical mapping and duplication events of *PRXs* in the *G. raimondii* genome. (D) Chromosomal physical mapping and duplication events of *PRXs* in the *G. arboreum* genome.

duplicated genes (GaPRX3:GaPRX7) on *G. arboreum* Ga_chr05 and Ga_chr08 was 23.98 MYA. The divergence time of these genes on Chr05 and Chr08 in he divergence time of *G. hirsutum* and *G. barbadense* was 23.2MYA. A comparative analysis of four cotton PRXs also showed that these segmental duplication genes were highly homologous, and the physical location of the PRXs in *G. arboreum* highly overlapped with that on the D subgenomes of *G. hirsutum* and *G. barbadense*.

## Analysis of *cis*-acting elements in promoter regions

In plants, *cis*-acting elements play an important role in gene expression regulation. We found many *cis*-acting elements in the promoters of the *PRX* genes in the four cotton species (Table S2) and identified three main types of *cis*-acting elements.

The first type of *cis*-acting element was responsive to abiotic stress (Fig. 5A). There were nine types of elements (ARE, DRE1, LTR, MBS, MYC, STRE, TC-rich repeats, WRE3, and WUN-motif) in this category, among which the most abundant *cis*-acting elements were related to drought stress. The elements related to drought stress were MBS and MYC, and MYC was more frequent. *GhPRX5-A*, *GhPRX6-A*, *GhPRX7-A*, *GhPRX9-D*, *GhPRX13-D*, *GbPRX1-A*, *GbPRX2-A*, *GbPRX5-A*, *GbPRX8-D*, *GbPRX9-D*, *GrPRX2*, *GrPRX8* and *GrPRX5* contained more than five drought-related elements among the four cotton species. Second in terms of presence, the *cis*-acting elements involved in the response to osmotic stress and defense were STRE- and TC-rich. In addition, there were elements that responded to anaerobic conditions, hypothermia and wounding.

The second type of *cis*-acting element was responsive to plant hormones (Fig. 5B). There were ten types of elements (ABRE, AuxRR-core, CGTCA-motif, ERE, GaRE-motif, P-box, TATC-box, TCA-element, TGaCG-motif, and TGa-element) that responded to abscisic acid, auxin, ethylene, gibberellin and SA. Among them, ABRE (ABA) and ERE were the most abundant in the cotton *PRX* gene promoters and were found in 31 and 36 *PRX* genes, respectively. We also found three types of elements (GaRE-motif, P-box and TATC-box) involved in the gibberellin response in the *PRX* promoters. Among them, there were 2 GaRE-motifs, 3 TATC-boxs and 14 P-boxs in 44 promoters. In addition, auxin response elements (AuxRR-core and TGA-element), methyl jasmonate response elements (MeJA and TGACG-motif), and SA response elements (TCA-element) were found in many *PRX* promoters.

The third type of *cis*-acting element was related to plant growth and development (Fig. 5C). There were nine such types of elements (AACA-motif, AT-rich element, CAT-box, circadian, GCN4-motif, HD-Zip 1, MBSI, MYB, and O2-site). Among them, only one circadian element was involved in circadian regulation, and this element was found in the promoters of only two *PRX* genes (*GhPRX10-D* and *GhPRX11-D*) in *G. hirsutum*. *PRXs* may be less sensitive to circadian rhythms. The largest number of *cis*-acting elements were MYB protein binding sequence sites, and two types of MYB protein sequence binding site elements (MBSI and MYB) were identified. In addition, some *cis*-acting elements (GCN4-motif and AACA-motif) involved in endosperm expression, *cis*-acting regulatory elements (CAT-box) involved in meristem expression, and response elements involved in palisade mesophyll cell differentiation (HD-Zip 1) were also found, and some light-related *cis*-acting elements were also identified (Fig. S5).

## Expression profiles of *PRX*s in *G. hirsutum* under stress

To investigate the regulation of *PRX* expression by *cis*-acting promoter elements, we analyzed the *cis*-acting elements that respond to drought (MBS and MYC), osmotic stress (STRE and TC-rich) and SA (TCA-element) in the promoter regions of *Gh-PRX* genes. The physical positions of these five *cis*-acting elements in the promoter regions of the *Gh-PRX*

| | A abiotic stress-responsive elements | | | | | | | | | B phytohormone responsive | | | | | | | | | | C plant growth and development | | | | | | | | |
|---|---|---|---|---|---|---|---|---|---|---|---|---|---|---|---|---|---|---|---|---|---|---|---|---|---|---|---|---|
| | ARE | DREI | LTR | MBS | MYC | STRE | TC-rich repeats | WRE3 | WUN-motif | ABRE | AuxRR-core | CGTCA-motif | ERE | GaRE-motif | P-box | TATC-box | TCA-element | TGaCG-motif | TGa-element | AACA_motif | AT-rich element | CAT-box | circadian | GCN4_motif | HD-Zip 1 | MBSI | MYB | O2-site |
| GhPRX1-A | 1 | 0 | 0 | 0 | 2 | 0 | 0 | 0 | 1 | 3 | 1 | 0 | 1 | 0 | 0 | 0 | 0 | 0 | 0 | 0 | 1 | 0 | 0 | 0 | 0 | 0 | 2 | 0 |
| GhPRX2-A | 0 | 0 | 0 | 0 | 2 | 0 | 0 | 1 | 0 | 1 | 0 | 1 | 1 | 1 | 0 | 1 | 1 | 1 | 0 | 0 | 0 | 1 | 0 | 1 | 0 | 0 | 2 | 0 |
| GhPRX3-A | 4 | 0 | 0 | 0 | 3 | 0 | 0 | 1 | 0 | 0 | 0 | 0 | 2 | 0 | 1 | 0 | 0 | 0 | 0 | 0 | 0 | 0 | 0 | 0 | 0 | 0 | 2 | 0 |
| GhPRX4-A | 0 | 0 | 0 | 1 | 1 | 1 | 0 | 0 | 1 | 2 | 0 | 0 | 2 | 0 | 0 | 0 | 1 | 0 | 0 | 0 | 0 | 1 | 0 | 1 | 0 | 1 | 1 | 0 |
| GhPRX5-A | 2 | 0 | 1 | 0 | 4 | 4 | 0 | 0 | 1 | 1 | 0 | 1 | 1 | 0 | 1 | 0 | 0 | 1 | 0 | 0 | 1 | 0 | 0 | 0 | 0 | 0 | 7 | 0 |
| GhPRX6-A | 1 | 0 | 0 | 0 | 4 | 1 | 0 | 2 | 1 | 0 | 2 | 1 | 0 | 0 | 1 | 0 | 2 | 1 | 1 | 0 | 0 | 0 | 0 | 0 | 1 | 0 | 2 | 0 |
| GhPRX7-A | 1 | 0 | 0 | 1 | 7 | 0 | 0 | 0 | 0 | 1 | 0 | 0 | 6 | 0 | 0 | 0 | 0 | 0 | 0 | 0 | 0 | 0 | 0 | 0 | 0 | 0 | 2 | 0 |
| GhPRX8-D | 1 | 0 | 0 | 0 | 3 | 0 | 0 | 0 | 0 | 1 | 0 | 1 | 4 | 0 | 0 | 0 | 1 | 1 | 0 | 0 | 0 | 0 | 0 | 0 | 0 | 0 | 1 | 0 |
| GhPRX9-D | 1 | 0 | 1 | 0 | 5 | 1 | 0 | 1 | 1 | 2 | 1 | 0 | 3 | 0 | 0 | 0 | 0 | 0 | 0 | 0 | 0 | 0 | 0 | 0 | 1 | 0 | 0 | 0 |
| GhPRX10-D | 0 | 0 | 0 | 2 | 1 | 1 | 0 | 1 | 1 | 3 | 0 | 0 | 1 | 0 | 0 | 0 | 0 | 0 | 1 | 0 | 0 | 0 | 1 | 1 | 0 | 0 | 3 | 0 |
| GhPRX11-D | 4 | 0 | 0 | 0 | 1 | 0 | 1 | 1 | 0 | 3 | 0 | 0 | 3 | 0 | 0 | 0 | 0 | 0 | 1 | 0 | 0 | 0 | 1 | 1 | 0 | 0 | 2 | 0 |
| GhPRX12-D | 0 | 0 | 0 | 1 | 2 | 1 | 0 | 0 | 0 | 3 | 0 | 1 | 3 | 0 | 0 | 0 | 0 | 1 | 1 | 0 | 0 | 0 | 0 | 1 | 0 | 0 | 2 | 0 |
| GhPRX13-D | 2 | 0 | 1 | 1 | 5 | 4 | 0 | 0 | 2 | 2 | 0 | 0 | 1 | 0 | 1 | 0 | 0 | 0 | 0 | 0 | 1 | 0 | 0 | 0 | 0 | 0 | 9 | 0 |
| GhPRX14-D | 1 | 0 | 0 | 0 | 3 | 0 | 1 | 0 | 1 | 3 | 0 | 0 | 1 | 0 | 0 | 0 | 0 | 0 | 1 | 0 | 0 | 0 | 0 | 1 | 0 | 0 | 2 | 0 |
| GhPRX15-D | 2 | 0 | 0 | 1 | 4 | 0 | 1 | 0 | 0 | 3 | 0 | 1 | 3 | 0 | 0 | 0 | 0 | 1 | 1 | 0 | 0 | 0 | 0 | 1 | 0 | 0 | 2 | 0 |
| GbPRX4-A | 1 | 0 | 0 | 0 | 2 | 0 | 0 | 0 | 1 | 2 | 0 | 0 | 2 | 0 | 0 | 0 | 1 | 0 | 0 | 0 | 0 | 1 | 0 | 1 | 0 | 1 | 2 | 0 |
| GbPRX6-A | 4 | 0 | 0 | 0 | 3 | 0 | 0 | 1 | 0 | 1 | 0 | 1 | 5 | 0 | 0 | 0 | 1 | 1 | 0 | 0 | 0 | 0 | 0 | 0 | 1 | 0 | 1 | 0 |
| GbPRX3-A | 0 | 0 | 0 | 1 | 1 | 0 | 0 | 0 | 0 | 3 | 1 | 0 | 1 | 0 | 0 | 0 | 0 | 0 | 0 | 0 | 1 | 0 | 0 | 0 | 0 | 0 | 3 | 0 |
| GbPRX2-A | 2 | 0 | 1 | 0 | 4 | 2 | 0 | 0 | 1 | 1 | 0 | 1 | 1 | 1 | 0 | 1 | 1 | 1 | 0 | 0 | 0 | 0 | 0 | 1 | 0 | 0 | 3 | 0 |
| GbPRX5-A | 1 | 0 | 0 | 0 | 3 | 2 | 0 | 2 | 1 | 0 | 0 | 1 | 1 | 0 | 1 | 0 | 0 | 1 | 0 | 1 | 0 | 0 | 0 | 0 | 0 | 0 | 2 | 0 |
| GbPRX1-A | 1 | 0 | 0 | 1 | 7 | 0 | 0 | 0 | 0 | 0 | 2 | 1 | 0 | 0 | 1 | 0 | 1 | 1 | 0 | 0 | 0 | 0 | 0 | 0 | 1 | 0 | 2 | 1 |
| GbPRX10-D | 1 | 0 | 0 | 0 | 3 | 1 | 0 | 0 | 0 | 3 | 0 | 0 | 1 | 0 | 0 | 0 | 1 | 0 | 0 | 0 | 0 | 0 | 0 | 1 | 0 | 1 | 1 | 0 |
| GbPRX8-D | 1 | 0 | 1 | 0 | 5 | 0 | 1 | 1 | 1 | 0 | 0 | 0 | 3 | 0 | 0 | 0 | 0 | 0 | 0 | 0 | 0 | 0 | 0 | 0 | 0 | 0 | 0 | 0 |
| GbPRX13-D | 1 | 0 | 0 | 1 | 2 | 1 | 0 | 0 | 1 | 2 | 1 | 0 | 4 | 0 | 0 | 0 | 0 | 0 | 0 | 0 | 0 | 0 | 0 | 0 | 0 | 0 | 3 | 0 |
| GbPRX11-D | 0 | 1 | 0 | 2 | 1 | 0 | 0 | 0 | 0 | 0 | 0 | 1 | 2 | 0 | 1 | 0 | 2 | 1 | 0 | 0 | 0 | 0 | 0 | 0 | 0 | 0 | 2 | 0 |
| GbPRX9-D | 2 | 0 | 1 | 1 | 3 | 2 | 0 | 0 | 3 | 1 | 0 | 0 | 6 | 0 | 0 | 0 | 0 | 0 | 0 | 0 | 0 | 0 | 0 | 0 | 0 | 0 | 2 | 0 |
| GbPRX7-D | 1 | 0 | 0 | 1 | 2 | 0 | 1 | 0 | 1 | 1 | 0 | 1 | 1 | 0 | 1 | 0 | 0 | 1 | 0 | 0 | 1 | 0 | 0 | 0 | 0 | 0 | 7 | 0 |
| GbPRX14-D | 2 | 0 | 0 | 1 | 4 | 0 | 1 | 0 | 0 | 3 | 0 | 0 | 2 | 0 | 0 | 0 | 0 | 0 | 0 | 0 | 0 | 2 | 0 | 1 | 0 | 1 | 1 | 0 |
| GaPRX5 | 1 | 0 | 0 | 0 | 0 | 0 | 0 | 0 | 1 | 0 | 0 | 1 | 1 | 0 | 1 | 0 | 0 | 1 | 0 | 1 | 0 | 0 | 0 | 0 | 0 | 0 | 2 | 0 |
| GaPRX4 | 0 | 0 | 0 | 1 | 0 | 0 | 0 | 1 | 1 | 0 | 2 | 1 | 0 | 0 | 1 | 0 | 1 | 1 | 0 | 0 | 0 | 0 | 0 | 0 | 1 | 0 | 1 | 1 |
| GaPRX3 | 1 | 0 | 0 | 2 | 0 | 1 | 0 | 1 | 2 | 1 | 0 | 0 | 0 | 0 | 0 | 0 | 1 | 0 | 0 | 0 | 0 | 0 | 1 | 1 | 1 | 0 | 2 | 0 |
| GaPRX1 | 4 | 0 | 0 | 0 | 0 | 0 | 0 | 1 | 0 | 0 | 0 | 0 | 3 | 0 | 0 | 0 | 0 | 0 | 0 | 0 | 0 | 0 | 0 | 0 | 0 | 0 | 4 | 0 |
| GaPRX7 | 0 | 0 | 0 | 1 | 0 | 1 | 0 | 0 | 1 | 2 | 1 | 0 | 4 | 0 | 0 | 0 | 0 | 0 | 0 | 0 | 0 | 0 | 0 | 0 | 0 | 0 | 3 | 0 |
| GaPRX6 | 2 | 0 | 1 | 0 | 0 | 4 | 0 | 0 | 1 | 0 | 0 | 1 | 1 | 0 | 1 | 0 | 2 | 1 | 0 | 0 | 0 | 0 | 0 | 0 | 0 | 0 | 2 | 0 |
| GaPRX8 | 1 | 0 | 0 | 1 | 0 | 0 | 0 | 0 | 1 | 1 | 0 | 0 | 6 | 0 | 0 | 0 | 0 | 0 | 0 | 0 | 0 | 0 | 0 | 0 | 0 | 0 | 2 | 0 |
| GaPRX2 | 1 | 0 | 0 | 0 | 0 | 1 | 0 | 2 | 0 | 0 | 2 | 1 | 0 | 0 | 0 | 0 | 1 | 1 | 1 | 0 | 0 | 0 | 0 | 0 | 1 | 0 | 1 | 0 |
| GrPRX6 | 1 | 0 | 0 | 0 | 4 | 0 | 0 | 0 | 0 | 0 | 0 | 1 | 1 | 0 | 1 | 0 | 0 | 1 | 0 | 1 | 0 | 0 | 0 | 0 | 0 | 0 | 2 | 0 |
| GrPRX7 | 0 | 1 | 0 | 1 | 1 | 2 | 0 | 0 | 0 | 0 | 0 | 0 | 1 | 0 | 1 | 0 | 0 | 0 | 0 | 0 | 1 | 0 | 0 | 0 | 0 | 0 | 8 | 0 |
| GrPRX8 | 2 | 0 | 1 | 1 | 3 | 2 | 0 | 0 | 3 | 2 | 2 | 1 | 0 | 0 | 0 | 0 | 1 | 1 | 0 | 0 | 0 | 0 | 0 | 0 | 1 | 0 | 1 | 1 |
| GrPRX3 | 2 | 1 | 0 | 1 | 2 | 0 | 0 | 1 | 1 | 3 | 0 | 0 | 0 | 0 | 0 | 0 | 1 | 0 | 0 | 0 | 0 | 0 | 0 | 1 | 0 | 1 | 2 | 0 |
| GrPRX1 | 3 | 0 | 0 | 1 | 2 | 0 | 2 | 1 | 0 | 2 | 1 | 0 | 5 | 0 | 0 | 0 | 0 | 0 | 0 | 0 | 0 | 0 | 0 | 0 | 0 | 0 | 3 | 0 |
| GrPRX2 | 1 | 0 | 1 | 0 | 6 | 0 | 1 | 1 | 0 | 0 | 0 | 0 | 2 | 0 | 0 | 0 | 0 | 0 | 0 | 0 | 0 | 0 | 0 | 0 | 0 | 0 | 3 | 0 |
| GrPRX4 | 1 | 0 | 1 | 0 | 4 | 0 | 1 | 0 | 1 | 1 | 0 | 1 | 0 | 0 | 1 | 0 | 0 | 1 | 0 | 0 | 0 | 0 | 0 | 0 | 0 | 0 | 3 | 0 |
| GrPRX5 | 2 | 0 | 0 | 0 | 5 | 0 | 2 | 0 | 0 | 3 | 0 | 1 | 3 | 0 | 0 | 1 | 0 | 1 | 1 | 0 | 0 | 0 | 0 | 1 | 0 | 0 | 1 | 0 |

**Figure 5  Cis-acting elements on the promoter of *PRX* genes in cotton.** (A) Abiotic stress-related cis-acting elements. (B) Plant hormone-induced cis-acting elements. (C) Cis-acting elements related to the growth and development of plants.

genes were plotted (Fig. 6). An element (MYC) involved in drought induction was present in all the *Gh-PRX* promoter regions, and the effects of drought on the expression levels of these genes were predicted to be strong.

We used PEG-6000, NaCl and SA to treat hydroponic *G. hirsutum* seedlings. Gene samples were extracted from *G. hirsutum* for qRT-PCR verification (Figs. 7 and 8). The expression level of genes in each time period was normalized to that at the 0 h time point. The results showed that nine *PRXs* were upregulated in plant roots under salt stress,

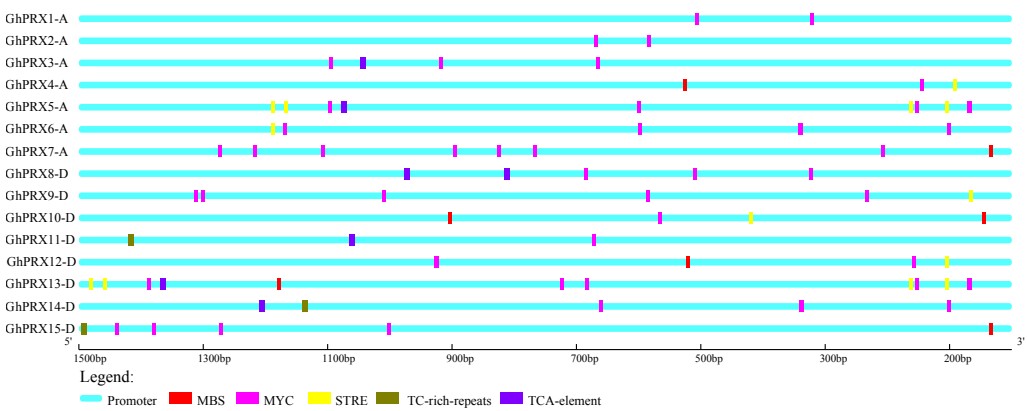

**Figure 6** **Analysis of cis-acting elements in the promoter region of *Gh-PRX* genes.** The promoter region has 1,500 bp before the gene translation initiation site.

including *GhPRX1-A*, *GhPRX2-A*, *GhPRX5-A*, *GhPRX10-D*, *GhPRX11-D*, *GhPRX12-D*, *GhPRX13-D*, *GhPRX14-D* and *GhPRX15-D* (Figs. 7A, 7B, 7E, 7J, 7K, 7L, 7M, 7N and 7O). The promoter regions of *GhPRX11-D*, *GhPRX14-D* and *GhPRX15-D* were rich in TC-rich elements, while *GhPRX5-A*, *GhPRX10-D*, *GhPRX12-D* and *GhPRX13-D* were rich in STRE elements. Some genes were repressed under stress for 1 h, such as *GhPRX4-A*, *GhPRX6-A*, *GhPRX8-A* and *GhPRX9-D* (Figs. 7D, 7F, 7H and 7I), in which *GhPRX4-A*, *GhPRX6-A* and *GhPRX9-D* were rich in STRE elements. However, in salt-treated leaves, the expression of most genes was inhibited, including *GhPRX2-A*, *GhPRX4-A*, *GhPRX5-A*, *GhPRX6-A*, *GhPRX8-D*, *GhPRX9-D*, *GhPRX10-D*, *GhPRX12-D* and *GhPRX13-D* (Figs. 8B, 8D, 8E, 8F, 8H, 8I, 8J, 8L and 8M), including the seven genes rich in STRE elements. Five genes were slightly upregulated under salt treatment, including *GhPRX1-A*, *GhPRX3-A*, *GhPRX11-D*, *GhPRX14-D* and *GhPRX15-D* (Figs. 8A, 8C, 8K, 8N and 8O). Among them, *GhPRX11-D*, *GhPRX14-D* and *GhPRX15-D* are rich in TC-rich elements. Although *GhPRX14-D* was slightly lower than the control at 0 h, it was higher than the expression of the control under stress. Under SA stress, five *PRXs* were downregulated in roots (*GhPRX1-A*, *GhPRX2-A*, *GhPRX4-A*, *GhPRX6-A* and *GhPRX8-D*) (Figs. 7A, 7B, 7D, 7F and 7H), seven *PRXs* were downregulated in leaves (*GhPRX2-A*, *GhPRX4-A*, *GhPRX5-A*, *GhPRX6-A*, *GhPRX8-D*, *GhPRX10-D* and *GhPRX12-D*) (Figs. 8B, 8D, 8E, 8F, 8H, 8J and 8L), and no significant upward trend was observed in all *PRXs*. Only *GhPRX8-D* in the root and *GhPRX13-D* in the leaf first decreased and then increased. *GhPRX2-A*, *GhPRX4-A*, *GhPRX6-A* and *GhPRX8-D* were rich in TCA-element *cis*-acting elements, and four genes were suppressed and downregulated in the roots and leaves.

MYC/MBS *cis*-acting elements exist in all *Gh-PRX* genes. Under drought stress simulated by PEG-6000, the expression of most *Gh-PRX* genes in roots first increased and then decreased (Fig. 7). Especially at 1 h, the expression levels of 10 of the 15 *PRX* genes increased significantly and then decreased rapidly (*GhPRX1-A*, *GhPRX2-A*, *GhPRX4-A*, *GhPRX5-A*, *GhPRX6-A*, *GhPRX8-A*, *GhPRX10-D*, *GhPRX11-D*, *GhPRX13-D* and *GhPRX15-D*) (Figs. 7A, 7B, 7D, 7E, 7F, 7H, 7J, 7K, 7M and 7O), which indicated that the *PRXs* in

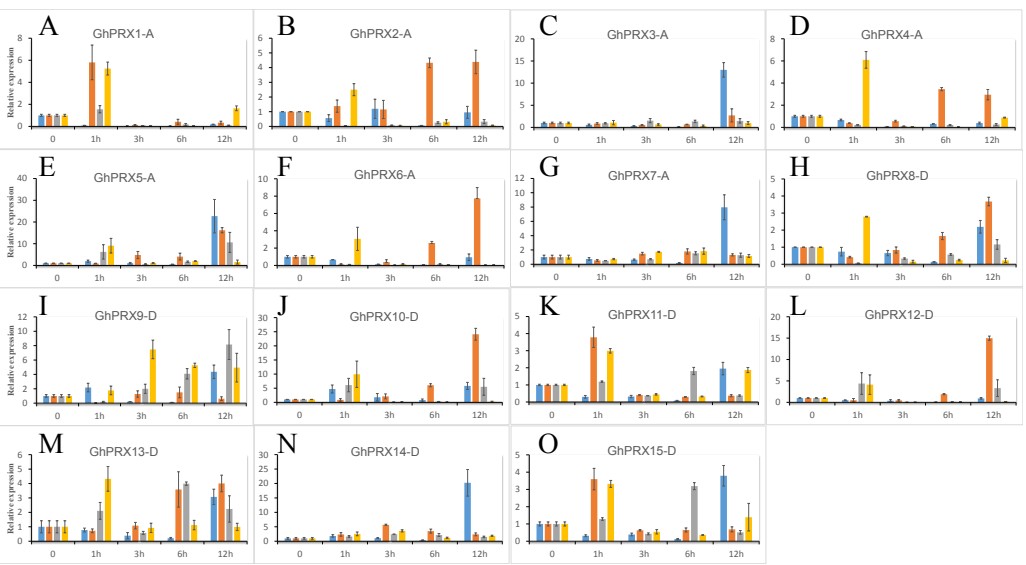

**Figure 7   Expression patterns of *Gh-PRX* genes in *G. hirsutum* roots.** (A–O) The expression patterns of each Gh-PRX member in *G. hirsutum* roots. The $2^{-\Delta CT}$ method was used to analyze the expression patterns of *Gh-PRX* genes in *G. hirsutum* roots. ($2^{-\Delta CT}$ calculation steps: 1. House-keeping gene homogenization sample difference: $\Delta CT = Ct$ target gene $-$ Ct house-keeping genes; 2. Comparison of stress samples and control samples at 0h: $\Delta Ct = \Delta Ct$ stress sample $-$ $\Delta Ct$ control sample; 3. Calculation: Fold change $= 2^{-\Delta CT}$). *G. hirsutum* seedlings were treated with drought, salt stress and SA hormone at the trefoil stage, and the expression patterns of *Gh-PRX* genes were identified by qRT-PCR. The blue strip column is the control, the orange strip column is NaCl stress, the gray bar column is SA stress, and the yellow bar column is drought stress. Using the cotton ubiquitin 7 gene as the house keeping gene, the statistical significant differences of the expression levels were show using the fold change values, all the values at all times were compared with the corresponding values at 0 h. Results are the average of three replicates, and the error bars indicating standard deviations.

roots were highly sensitive to drought stress and induced. However, in leaves, most *PRXs* showed a downregulation trend, such as *GhPRX2-A*, *GhPRX4-A*, *GhPRX5-A*, *GhPRX6-A*, *GhPRX8-D*, *GhPRX10-D* and *GhPRX12-D*. Only *GhPRX9-D* showed the same expression as roots, first upregulated and then downregulated (Fig. 8I).

## Expression and AS profiles of *PRX*s in different tissues of *G. hirsutum*

It is important to analyze the expression of genes in plant tissues and organs to understand gene function. We analyzed the available transcriptome data of various tissues and organs of *G. hirsutum* TM-1 (*Zhang et al., 2015*), including the roots, stems, leaves, cotyledons, petals, stamens, ovules, seeds, and fibers at 5, 10, 15 and 20 DPA. A heatmap was used to show the expression of the *PRX* genes (Fig. 9A). *GhPRX3-A*, *GhPRX5-A*, *GhPRX10-D*, *GhPRX11-D* and *GhPRX13-D* were expressed at low levels or not expressed in all tissues. *GhPRX3* and *GhPRX12* were highly expressed in all tissues, and the FPKM value of GhPRX14 expression level in roots and leaves was approximately 50–60.

Understanding and verifying the AS events of the *G. hirsutum PRX* family is an important step in the study of gene functional differentiation. Five main types of AS events were used
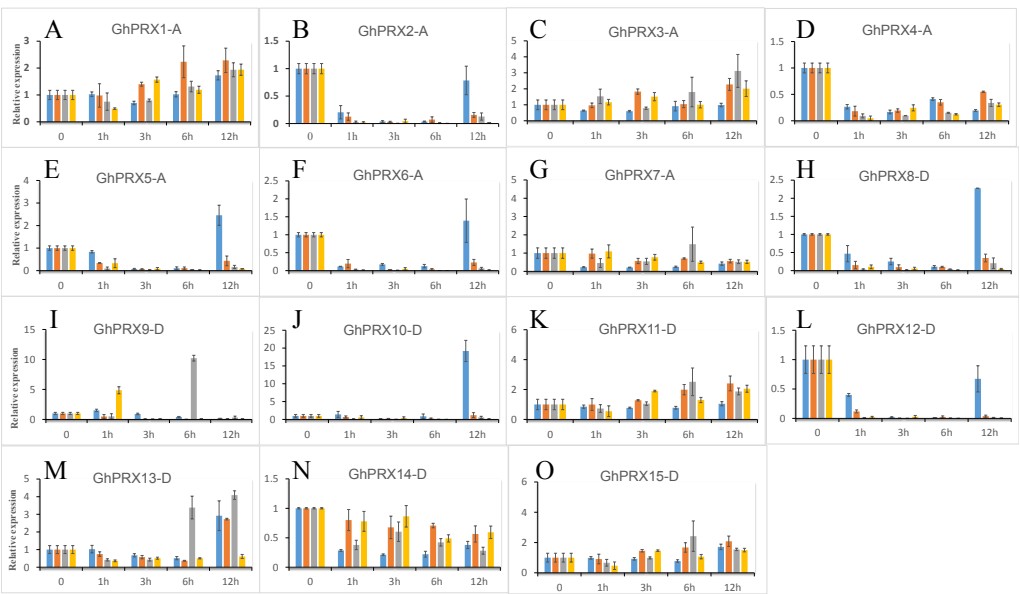

**Figure 8  Expression patterns of *Gh-PRX* genes in *G. hirsutum* leaves.** (A–O) The expression patterns of each Gh-PRX member in *G. hirsutum* leaves. The $2^{-\Delta CT}$ method was used to analyze the expression patterns of *Gh-PRX* genes in *G. hirsutum* leaves. ($2^{-\Delta CT}$ calculation steps: 1. House-keeping gene homogenization sample difference: $\Delta CT = Ct$ target gene $-Ct$ house-keeping genes; 2. Comparison of stress samples and control samples at 0 h: $\Delta Ct = \Delta Ct$ stress sample $-\Delta Ct$ control sample; 3. Calculation: Fold change $= 2^{-\Delta CT}$). *G. hirsutum* seedlings were treated with drought, salt stress and SA hormone at the trefoil stage, and the expression patterns of *Gh-PRX* genes were identified by qRT-PCR. The blue strip column is the control, the orange strip column is NaCl stress, the gray bar column is SA stress, and the yellow bar column is drought stress. Using the cotton ubiquitin 7 gene as House-keeping gene, the statistical significant differences of the expression levels were show using the fold change values, all the values at all times were compared with the corresponding values at 0 h. Results are the average of three replicates, and the error bars indicating standard deviations.

for further analysis: exon skipping (ES), intron retention (IR), 5′ or 3′ alternative splice sites (A5SS or A3SS), alternative first exon (AFE), and alternative last exon (ALE). Because change in the CDS can modify protein sequence and function, only AS events located in the CDSs of the genes were used for further analysis. Using the full-length primers designed for the amplification of *PRX* gene cDNA, RT-PCR amplification was performed using the leaf and root cDNA of *G. hirsutum* sGK9708 as a template. The results showed that the products obtained for *GhPRX14-D* included multiple fragments (Figs. 9B, 9C and 9E); moreover, s everal bands with different levels of brightness and unequal size were present. Thus, we speculate that *GhPRX14-D* may have multiple transcripts. The PCR products of *GhPRX14-D* were recovered, purified and ligated into a cloning vector for replication and sequencing. Analysis of the obtained sequences showed two different transcripts of *GhPRX14-D* in the leaves of *G. hirsutum* (Table 2) and three in the roots. *GhPRX14-D* uses three AS methods, IR, ES and A3SS. *GhPRX14-Leaf-AS2* and *GhPRX14-Root-AS2* use one nonstandard AS site: 5′-AG. AA-3′ (3 clones and 2 clones), which retains 15 bases at the 3′ end of the first intron (Fig. 9D). *GhPRX14-Root-AS3* uses two nonstandard AS sites:
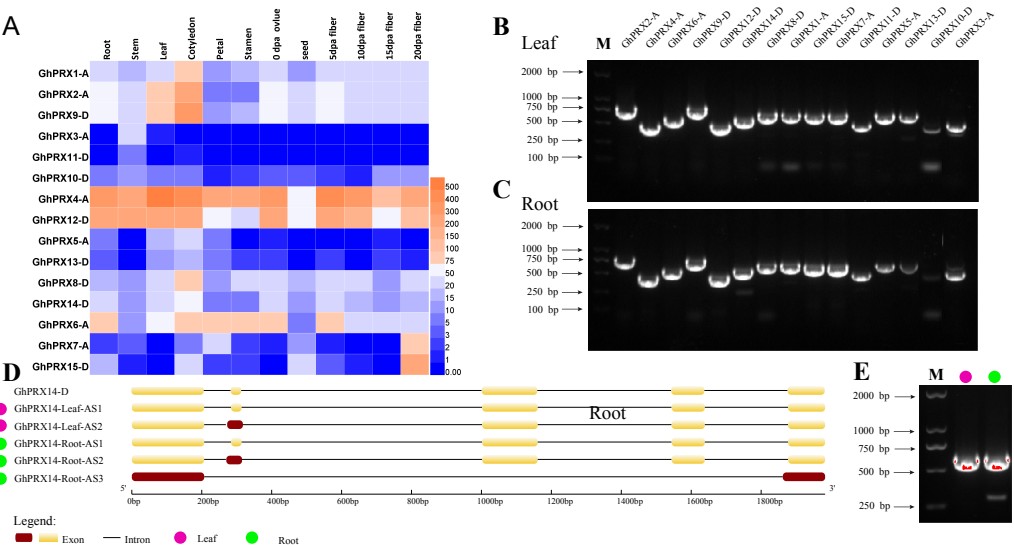

**Figure 9** **Analysis of tissue expression and alternative shearing profiles of *PRX* genes.** (A) Expression patterns of *PRXs* in different tissues and organs of *G. hirsutum* TM-1. The color represents *Gh-PRX* expression levels: Log2 (FPKM). (B) Agarose gel electrophoresis map of *PRXs* in *G. hirsutum* sGK9708 leaves. (C) Agarose gel electrophoresis map of *PRXs* in *G. hirsutum* roots. (D) Structural diagram of *GhPRX14* alternative shearing transcripts. *GhPRX14-D* is the reference transcript of *G. hirsutum* TM-1. The DNA marker is the Trans2K DNA marker from TransGen. (E) Agarose gel electrophoresis map of *GhPRX14-D* in *G. hirsutum* leaves and roots.

**Table 2** **Alternative splicing profile analysis of *PRXs*.**

| Transcript | Alternative splicing mode | Alternative splicing site | Alternative splicing boundary | Number of clones | ORF length (bp) | Amino acid length (aa) |
|---|---|---|---|---|---|---|
| GhPRX14-Leaf-AS1 | – | – | – | 24 | 597 | 199 |
| GhPRX14-Leaf-AS2 | IR | The first intron 3′ 15-bp base | AG/AA | 3 | 612 | 204 |
| GhPRX14-Root-AS1 | – | – | – | 24 | 597 | 199 |
| GhPRX14-Root-AS2 | IR | The first intron 3′ 15-bp base | AG/AA | 2 | 612 | 204 |
| GhPRX14-Root-AS3 | ES and A3SS | The fourth intron 3′ 14-bp base, the second, third, and fourth exons were skipped. | AG/AT, GG/TT | 2 | 327 | 109 |

5′-AG. AT-3′ (2 clones) and 5′-GG. TT-3′ (2 clones), which retain 14 bases at the 3′ end of the fourth intron. The second, third, and fourth exons were skipped (Fig. 9D).

## Subcellular localization of *PRX* proteins

Determining the distribution of proteins in cells is an important step to verify protein function. Using the TargetP 1.1 Server (http://www.cbs.dtu.dk/services/TargetP/) website to predict and analyze proteins, we found that the *PRX* proteins were mainly located in
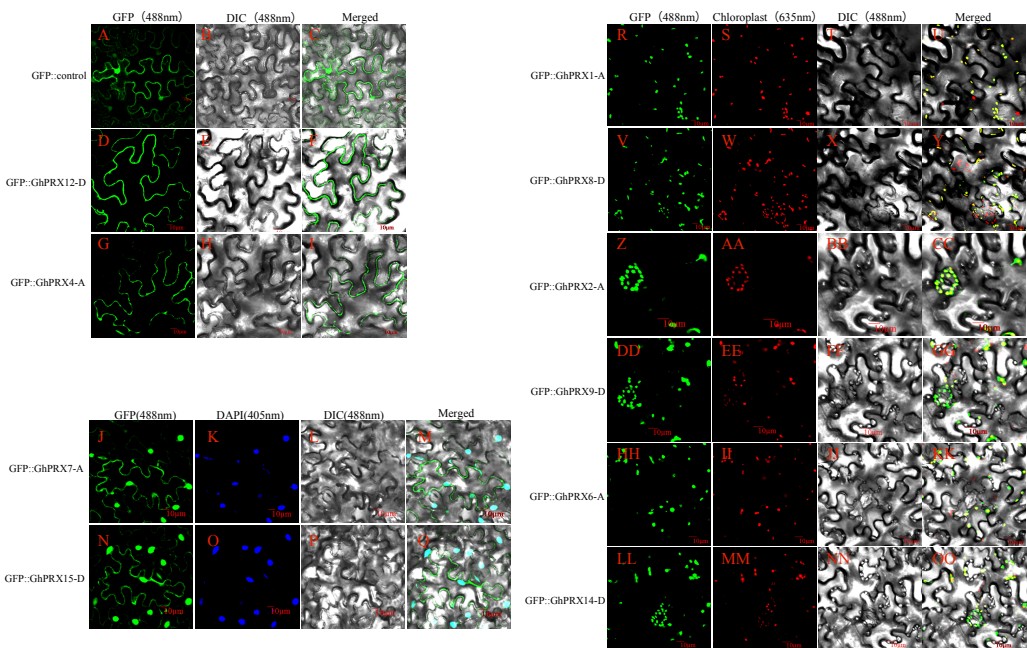

**Figure 10 Subcellular localization map of *PRX* proteins.** Subcellular localization map of *PRX* proteins. (A–I) The protein fluorescence signals of the eGFP control, *GhPRX4-A* and *GhPRX12-D* are located on the cell membrane; (J–Q) The protein fluorescence signals of *GhPRX7-A* and *GhPRX15-D* are located on the cell membrane and nucleus; (R-OO) The protein fluorescence signals of *GhPRX1-A*, *GhPRX8-D*, *GhPRX2-A*, *GhPRX9-D*, *GhPRX6-A* and *GhPRX14-D* are located on chloroplasts. GFP is a fluorescence signal excited by recombinant pcambia2300-eGFP (CAMBIA) transient expression vector, DAPI stains the nucleus, and chloroplast autofluorescence was obtained at 635 nm fluorescence receiving wavelength.

chloroplasts and plasma membranes. However, the specificity of a few gene prediction results was too low to determine protein localization (Table S4).

To verify the predicted subcellular localizations of the *PRX* proteins, we selected 10 *PRXs* with high expression levels for subcellular localization analysis (5 pairs of homologous genes). We constructed a transient expression 35s-PRX-eGFP vector for each gene that was used to verify the experimental purpose. The results showed that the eGFP fluorescence signal in the control group was dispersed throughout the tobacco leaf cells, while the fluorescence signals of *GhPRX4-A* and *GhPRX12-D*, which are members of PRXIIB, were located in the cell membrane (Figs. 10D–10I), thus providing support for the accurate classification of PRXIIB subfamily members. The protein fluorescence signals of *GhPRX7-A* and *GhPRX15-D*, members of 1-CysPRX, were located on the cell membrane and nucleus (Figs. 10J–10M and 10N–10Q). *GhPRX1-A* and *GhPRX8-D* of PRXIIE, *GhPRX6-A* and *GhPRX14-D* of PRXIIF, *GhPRX2-A* and *GhPRX9-D* of 2-CysPRX, and their protein fluorescence signals were located in chloroplasts (Figs. 10R–10OO).

## DISCUSSION

*PRX* is a ubiquitous and abundant protein and is very important for oxidation resistance and cell signal transduction. Recently, in plants, 2-CysPRX in chloroplasts was shown

to regulate protein redox status and oscillate diurnally between hyperoxidation and reduction, thus protecting plants from a myriad of harsh environmental stresses (*Lee et al., 2017*) and further protecting prochlorophyllide synthesis, which takes place through the stimulation of an aerobic cyclase by the NTRC/2-CysPRX system (*Stenbaek et al., 2008*). 2-CysPRX deficiency inhibits photosynthesis and chlorophyll accumulation during plant development (*Baier & Dietz, 1999*). 2-CysPRX proteins in plants are involved in key signaling processes, such as WWC and H2O2-mediated signaling in plastids (*Awad et al., 2015*) and carbon metabolism (*Cerveau et al., 2016*), and they are presumed to act as plastidial thiol oxidases (*Cerveau et al., 2019*). These recent advances in understanding 2-CysPRX functions indicate that this family of genes is involved in the photosynthesis and antioxidation pathways in chloroplasts, and our subcellular localization in this study confirms that 2-CysPRX genes are mainly distributed in cotton chloroplasts (Figs. 10R–10OO). PRXQ represents approximately 0.3% of chloroplast proteins, and is localized on the thylakoid membrane. Data show that PRXQ is directly involved in photosynthesis (*Lamkemeyer et al., 2006*). *PRX* type II exists in the cytoplasm, plastid and mitochondria as general antioxidant active proteins. In addition, type II *PRX* proteins protect DNA from damage in vivo or in vitro (*Banmeyer et al., 2005*). 1-CysPRX proteins are targeted to the nucleus and cytosol in higher plants and related to seed dormancy and germination (*Goldmark et al., 1992*; *Haslekås et al., 2003*). However, a genome-wide identification or evolutionary analysis of the *PRX* gene family has not been reported in cotton, although four cotton species have been sequenced. Here, we identified 15, 9, 8 and 13 *PRXs* from *G. hirsutum*, *G. raimondii*, *G. arboreum* and *G. barbadense*, respectively. All cotton *PRX* protein sequences were aligned with the NCBI reference protein (refseq_protein) database. We found that the study of this family of proteins in cotton was still in the prediction stage (Table S5) and that some predicted subclassifications were not entirely accurate. In terms of protein similarity, the sequences of the cotton *PRX* family proteins were most similar to those of *Durio zibethinus* and *Citrus sinensis*.

We used the DASP method to classify the cotton *PRX* proteins into subfamilies. The correct subfamily classification of the *PRX* family has remained difficult because *PRX* subfamily identity is usually independent of phylogenetic distribution; thus, the subfamilies do not correspond to the phylogenetic classification based on the PSI-BLAST method (*Dietz, 2011*; *Soito et al., 2011*). Using the DASP method to classify proteins according to their conserved active functional sites can provide a more accurate subfamily classification (*Nelson et al., 2011*). We found that all cotton *PRX* proteins contained a completely conserved active site cysteine (PXXXTXXCp), called peroxidatic cysteine (Cys$_P$-SH), which is the catalytic center of the peroxidase. The thiolate of Cys$_P$-SH attacks the peroxide substrate (*Dietz, 2011*), reducing the peroxide to alcohol and water or nitrite (*Hofmann, Hecht & Flohé, 2002*; *König et al., 2002*). *PRX* subfamily classification is based on conserved active site sequence information (PxxxTxxC…S…W/F) and gene structural similarity. To clarify the classification, we also referred to the traditional subclassification based on the positions of the conserved cysteinyl residues. The results showed that 39 *PRX* proteins were correctly classified in cotton using the DASP method, and the sequences of the 5 incorrectly classified genes (*GhPRX3-A*, *GhPRX11-D*, *GbPRX6-A*, *GrPRX1* and *GaPRX1*) were highly
similar to the sequences of PRXIIB subfamily proteins with the second cysteine residue replaced (Fig. 1). Thus, these proteins should be classified as 1-CysPRX, although the currently available predictions for these five genes (*GhPRX3-A*, *GhPRX11-D*, *GbPRX6-A*, *GrPRX1* and *GaPRX1*) still list them as PRXIIB (accession numbers: XP_016702396.1, XP_016684751.1, XP_016724513.1, XP_012440686.1, XP_017605848.1). Tandem gene duplication, segmental duplication and polyploid events are the main means of gene family expansion (*Cannon et al., 2004*). Thus, gene duplication events must be analyzed at the genome level. Our comparative genome analysis also showed that a large number of *PRXs* are produced by segmental duplication and polyploid events, which greatly enriches the members of the *PRX* family. At present, *G. raimondii* (D5 genome) is generally considered to be the D subgenome donor of *G. hirsutum* (*Li et al., 2015*). However, a comparative analysis of the physical location and sequence of *PRX* family genes in four cotton species showed that the gene information of *PRXs* in *G. arboreum* is more homologous to that in *G. hirsutum* and *G. barbadense*, which was verified by both the location information on the chromosome and the time of homologous genes divergence. Therefore, we speculate that the *PRXs* of diploid *G. arboreum* are the donors of *PRXs* in the D subgenomes of allotetraploid *G. hirsutum* and *G. barbadense*.

To understand the expression patterns of *PRXs* during cotton growth and development, the effect of stress on gene expression must be investigated. We found abundant *cis*-acting elements related to abiotic stress, hormone induction and plant growth and development regulation in the promoter regions of these genes in cotton (Fig. 5), suggesting that *PRX* genes are widely involved in the growth and development of cotton plants. The qRT-PCR analysis showed that most *PRXs* could be induced or repressed by abiotic stress and hormone treatments, indicating that *PRXs* may play important roles in coping with stress. Many studies have shown that the expression level of *PRXs* respond to drought (*Cho et al., 2012*; *Haddad & Japelaghi, 2015*; *Xu et al., 2015*; *Xu et al., 2019*), abscisic acid (*Baier, Ströher & Dietz, 2004*; *Haslekås et al., 2003*), and ethylene stress (*Tovar-Méndez et al., 2011*); moreover, Arabidopsis overexpressing *PRX* shows increased salt and low temperature tolerance (*Jing et al., 2006*). Based on these stresse-induced changes in *PRX* expression, we aimed to understand the role of *cis*-acting elements in this regulation. Our study showed that expression of three genes (*GhPRX11-D*, *GhPRX14-D* and *GhPRX15-D*) rich in TC+rich elements was induced and three of the seven genes with STRE elements (*GhPRX4-A*, *GhPRX6-A* and *GhPRX9-D*) were repressed in roots and leaves under salt stress (Figs. 7 and 8). In some existing studies, *PRX* RNA and protein levels in mung beans decreased under high salinity (*Cho et al., 2012*) while *PRX* protein expression in rice (*Oryza sativa* L.) was upregulated under high salinity (*Xu et al., 2015*). These findings indicate that *PRXs* are differentially expressed under salt stress, and TC-rich and STRE elements may play certain roles in the regulation of *PRX* expression. Under treatment with SA hormone, the expression of most *PRXs* is repressed. Studies have shown that SA can greatly alleviate oxidative damage under stress by activate the mechanism of scavenging reactive oxygen species and reducing the activity of antioxidant enzymes (*Soltani Delroba, Karamian & Ranjbar, 2011*), which may also lead to the inhibition of *PRX* expression. We found that the expression of all *PRXs* rich in TCA-elements in cotton was repressed

(Figs. 7B, 7D, 7F, 7H and 8B, 8D, 8F, 8H). To date, studies of TCA-elements in plants have shown that these elements can be activated by SA (*Salazar et al., 2007*). Therefore, we speculate that TCA-elements play an inhibitory role in *PRX* expression in cotton, which provides a reference for the regulation of genes by this element in other plants. Under drought stress, the expression of 10 of 15 *PRXs* in *G. hirsutum* roots increased rapidly at 1 h and then decreased gradually (Figs. 7A, 7B, 7D, 7E, 7F, 7H, 7J, 7K, 7M and 7O). In leaves, only GhPRX9-D was upregulated and then downregulated, while the other *PRXs* did not fluctuate or decreased slowly within 12 h (Fig. 8). Previous studies in drought-stressed alfalfa showed that *PRXs* were repressed in the shoot but induced in the root (*Kang & Udvardi, 2012*), roots are highly sensitive to drought, which increases the production of intracellular reactive oxygen species (ROS) (*Hu et al., 2011*). Therefore, roots directly exposed to drought stress will rapidly upregulate the expression of *PRXs* to regulate the level of ROS (*Zhou et al., 2014*). A decrease in the water potential in leaves may not be sufficient to induce the expression of *PRXs* and downregulate the expression of *PRXs* in leaves (*Bhardwaj, Mala & Kumar, 2014*). Since MYC and MBS elements are present in all *PRX* promoters, the role of MYC and MBS elements in regulating *PRX* expression in different *G. hirsutum* tissues has not been determined. Since these *cis*-acting elements respond to stress, the verification of regulated gene expression depends on additional research data to identify the transcription factors that interact with these elements.

AS is a key posttranscriptional regulatory mechanism that can produce multiple transcripts and protein isomers, enrich protein diversity and increase protein functional complexity. Studies have shown that AS is an important form of plant gene expression regulation (*Syed et al., 2012*) that is involved in many physiological metabolic processes, signal transduction and responses to external biotic and abiotic stresses in plants (*Barbazuk, Fu & McGinnis, 2008*; *Mastrangelo et al., 2012*). A large amount of data has shown that IR is the most common AS event (*Campbell et al., 2006*; *Rauch et al., 2013*; *Reddy et al., 2013*; *Zhiguo, Wang & Zhou, 2013*) and ES is a less common event (*Barbazuk, Fu & McGinnis, 2008*). Our analysis of the AS events of *PRXs* in *G. hirsutum* showed that the main AS modes were IR, ES, and A5SS or A3SS (Table 2). By examining the type and location of AS, we found that *GhPRX14-D* has the same type of transcript in plant roots and leaves, *GhPRX14-Leaf-AS2* and *GhPRX14-Root-AS2* (Figs. 9D and 9E). In addition, the specific transcript *GhPRX14-Root-AS3* has two AS types, ES and A3SS. The existence of two or more AS types for each transcript as well as differential splicing of homologous genes are very common in plants (*Chen et al., 2018*; *Zhang et al., 2019*). We found that there were no AS events in *GhPRX6-A,* the homologous gene of *GhPRX14-D*, which indicated that the homologous gene was differentially expressed with respect to AS events. AS transcripts use two or more AS methods, and the differential splicing of homologous genes significantly increases transcript complexity and enriches protein types. Therefore, verifying the AS of *PRXs* can help us understand the distribution of the transcripts of this gene family in cotton and provide a valuable reference for future studies of the diversity of this protein family in cotton.

Determining the distribution of proteins in cells is an important step for analyzing the subclassification and protein function of the PRX family. In this study, 10 PRXs
with the highest expression level were selected, and 4 PRX proteins were found to be located in chloroplasts (Figs. 10Z–10GG and 10R–10Y), and they belonged to the 2-CysPRX and PRXIIE subfamilies. In many other species, the two subfamily proteins have also been proven to be mainly located in chloroplasts (*Baier & Dietz, 1997*). They mainly function in chloroplasts, including participating in the balance of the thiol redox system of chloroplasts (*Pérez-Ruiz et al., 2017*), metabolism regulation (*Yoshida et al., 2018*), and signal transduction (*Awad et al., 2015*). In the transient expression assay in onion epidermal cells, a green fluorescent protein-AtPER1 (1-CysPRX) fusion protein was also localized to the cytoplasm and nucleus. The localization information of cotton 1-CysPRX proteins also demonstrated that it was mainly located in the nucleus and cytoplasmic membrane (Figs. 10J–10Q). Strangely, previous studies showed that PRXIIF is mainly located in mitochondria (*Horling, König & Dietz, 2002*; *Iglesias Baena, 2010*) and decomposes peroxides in the presence of glutaredoxins as a reductant. However, our localization of PRXIIF proteins shows that it is also located in chloroplasts. Conservative sequence alignment of PRXII proteins shows that the conserved S (serine) is mutated into A (alanine), which may be characteristic of the PRXIIF sequence in chloroplasts. These results provide new research data for PRXII subfamily classification and localization.

## CONCLUSIONS

In this study, to thoroughly characterize the *PRX* family and correct the current misclassification of *PRXs* in 4 cotton species, we identified 15, 8, 8 and 13 *PRX* genes from *Gossypium hirsutum*, *G. raimondii*, *G. arboreum* and *G. barbadense*, respectively. Forty-four *PRXs* were identified and divided into six subfamilies, namely, 2-CysPRX, 1-CysPRX, PRQ, PRXIIB, PRXIIE and PRXIIF. The distribution, evolution, regulation mechanism, alternative splicing and location of these *PRXs* in cotton were analyzed and identified in detail. This study provides the first systematic report on *PRXs* in cotton. It aims to provide a foundation for further research on biological resistance to stress, anti-aging and disease resistance.

**Abbreviations List**

| | |
|---|---|
| **PRX** | Peroxiredoxin |
| **SA** | salicylic acid |
| **IR** | intron retention |
| **ES** | exon skipping |
| **A5SS or A3SS** | 5′ or 3′ alternative splice sites |
| **AS** | alternative splicing |
| **pI** | isoelectric point |
| **MW** | molecular weight |
| **FPKM** | Fragments Per Kilobase of exon model per Million mapped fragments |

## ACKNOWLEDGEMENTS

We would like to thank Peng Huo (The Institute of Cotton Research, Chinese Academy of Agricultural Sciences) for assistance with laser confocal microscopy (FV1200, OLYMPUS).

### Funding

This work was funded by the Natural Science Foundation of China (No. 31621005 and 31471538). Central Public interest Scientific Institution Basal Research Fund (No.1610162019010101), the State Key Laboratory of Cotton Biology Open Fund (CB2017A05) and the Agricultural Science and Technology Innovation Program for CAAS (CAASASTIPICRCAAS). The funders had no role in study design, data collection and analysis, decision to publish, or preparation of the manuscript.

### Grant Disclosures

The following grant information was disclosed by the authors:
Natural Science Foundation of China: 31621005, 31471538.
Central Public interest Scientific Institution Basal Research Fund: 1610162019010101.
State Key Laboratory of Cotton Biology Open Fund: CB2017A05.
Agricultural Science and Technology Innovation Program for CAAS (CAASASTIPICR-CAAS).

### Competing Interests

The authors declare there are no competing interests.

### Author Contributions

- Yulong Feng and Renhui Wei conceived and designed the experiments, performed the experiments, analyzed the data, prepared figures and/or tables, authored or reviewed drafts of the paper, and approved the final draft.
- Aiying Liu, Senmiao Fan, Zhen Zhang and Baoming Tian analyzed the data, authored or reviewed drafts of the paper, and approved the final draft.
- JinCan Che performed the experiments, authored or reviewed drafts of the paper, and approved the final draft.
- Youlu Yuan, Gongyao Shi and Haihong Shang conceived and designed the experiments, authored or reviewed drafts of the paper, and approved the final draft.

### Data Availability

  Raw data are available in the Supplemental Files.

### Supplemental Information

Supplemental information for this article can be found online at http://dx.doi.org/10.7717/peerj.10685#supplemental-information.

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
