# Peer review of "Genome-wide identification, evolution, expression, and alternative splicing profiles of peroxiredoxin genes in cotton"

_PeerJ, doi:10.7717/peerj.10685_

## Round 0.1 · original submission · Major Revisions

All reviewers made positive comments on your manuscript. However, the present version of the manuscript is quite hard to read, even for scientists familiar with the field of plant PRXs. Please pay attention to the comments, and address the following concerns when you make revision: i) the writing style, ii) an inappropriate classification of PRXs and iii) too little consideration in the bibliographical context. In other respects, the expression data and their interpretation are highly questionable.

Reviewer 1 ·

Basic reporting

Peroxiredoxins (PRXs) are ubiquitous thioredoxin-dependent peroxidases that have been relatively well characterized in plants over the past 25 years. They display some diversity regarding redox-active Cys and catalytic mechanisms, leading to a recognized classification in the plant kingdom (see reviews by Dietz 2011, ARS, and by Liebthal et al., 2018, ARS). They fulfil key roles in the maintenance of redox homeostasis via the scavenging of hydrogen peroxide, but also in the direct transmission of redox signals to protein partners as shown by the key findings published in the last years.

The manuscript by Feng et al. aims to characterize in a thorough way the PRX families in 4 cotton species, in which these proteins have hardly been studied. Thus, the submitted work is original and contains a huge amount of data related to sequence analysis, expression patterns and subcellular localization. Some, such as those related to alternative splicing, are novel and interesting. Therefore, this work could be a valuable contribution to the field. However, the present version of the manuscript is quite hard to read, even for scientists familiar with the field of plant PRXs. This is notably due to: i) the writing style, ii) an inappropriate classification of PRXs and iii) too little consideration in the bibliographical context. In other respects, the expression data and their interpretation are highly questionable.

Experimental design

The research plan is globally well designed.

Validity of the findings

- 1) The proposed classification of PRX genes in Figure 1 is not appropriate and very confusing for people working on plant PRXs. Further, it does not even correspond to the one used for animal PRXs (Perkins et al. 2015, TIBS). The authors have to take into consideration the extensive work performed by Prof. Dietz’s group and published in several reviews and articles (Dietz, 2011, ARS; Liebthal et al., 2018, ARS). TPX, PRX2, PRX5, PRX1 and PRX6 are in fact PRXIIB, PRXIIE, PRXIIF, 2-CysPRX and 1-CysPRX. This classification is commonly used in the plant field.

- 2) Similarly, the classification proposed by the authors in Fig. 3 (PTC2 and PTC4, lines 241-265) should be reconsidered in the context of what is already known on plant PRXs.

- 3) The expression data (presented in Figures 7 and 8, the quality of which is poor) are highly questionable. The authors have to take care in their interpretation. How do they explain the highly increased expression of several PRX genes 12 h in control conditions? This is very surprising and calls into question the validity of their findings since the control levels should be the same throughout the experience. This is not the case for many genes.
Numerous interpretations are incorrect, for instance, the authors claim lines 373-374 that salt induces the expression in leaves of PRX11D and PRX15D, but when looking at Fig. 8, the increases are very weak.

Additional comments

- 1) The cited bibliography regarding plant PRXs is too limited and not up to date. Indeed, in the Introduction, the authors put relatively more emphasis on animal PRXs than on plant PRXs. Further, important findings regarding plant 2-CysPRX functions, notably in signaling pathways, that have been obtained in the last years are not cited (Awad et al, 2015, Plant Phys; Cerveau et al., 2016, Plant Sci; Yoshida et al., 2018, PNAS; Vaseghi et al., 2018, eLife; Cerveau et al., JExpBot, 2019…). The references cited are not always accurate. For instance, Cerveau et al., line 468 is not the right one for photosynthesis deactivation. Banmeyer et al, (2005) (line 472) is not present in the reference list. Conversely, Cerveau et al., (2016) is present in the list (line 617) but not cited in the text.

- 2) The manuscript is rather hard to read due to the poor writing style and too many awkward sentences (for instance, “…all PRX genes increased…” lines 546-547).Thus, I recommend the authors to have their manuscript edited by a native English speaker. In addition, the high number of genes cited makes the style very confused (see paragraph from line 291 to line 317) or vey repetitive in some parts (see lines 411-419, where five consecutive sentences are constructed in the same way). The authors should highlight the main results in a more synthetic way.

- 3) The Discussion has to be thoroughly improved. The present version does not take sufficient account of bibliographical data (see first comment). The paragraph from line 500 to line 519 again comments on the results

Minor comments:
- 4) What does mean “PRX protease” lines 46-47?
- 5) Line 331: Suppress the last sentence.
- 6) Lines 463-464: Suppress the title of the reference.

·

Basic reporting

The paper by Feng et al. identified a total of 47PRX genes in the cotton genome. The authors showed that the PRX gene family is divided into six subfamilies according to the phylogenetical analysis of the conserved active site. Furthermore, they also confirmed that cis acting elements could effectively regulate the expression of PRX genes. Based on the results presented in this paper, most PRX members were situated in chloroplasts, and a few members were found in the cell membrane and nucleus.

The language of the paper could be improved a bit if reviewed by a native speaker. The introduction provides a good, generalized background of the topic that quickly gives the reader an appreciation of the wide range of applications for this experiment. The title and abstract are appropriate for the content of the text. Furthermore, the article is well constructed, the experiments were well conducted, and analysis was well performed. Almost in all cases I could find that the references are relevant. The literature cited is relevant to the study, but there are several instances, which have been noted in the text, in which the author makes assertions without substantiating them with references.

Experimental design

The experimental apparatus is quite standard, and is appropriate for the study. I don’t think any additional experiments are necessary to validate the results presented here, However, I believe the authors should add some more information to the materials and method section as some information are missing in the current format. The authors need to improve the clarity of the methodology section by bringing more information on how they did the experimental parts. For example, when comes to phylogenetical analysis, the authors are not precise on how they did the phylogeny, what was the bootstrap value they used, what was the model they used for their phylogeny, and so on. In another example, when the author talks on RT-PCR analysis it’s not clear which statistical analysis used by authors? Please also see my comments on the text.

Validity of the findings

All data to support the idea behind this work have been provided. However, the authors should bring more information on the statistical analysis of their RT-PCR data as suggested earlier. The conclusion part needs to be written better than the current format. You should develop the conclusion by briefly restating the rationale for your experiment and the purpose of the article, then discussing the conclusions you have drawn. You should also discuss the implications of your experimental findings and where you think research in this field should go from here.

Additional comments

The paper by Feng et al. identified a total of 47PRX genes in the cotton genome. The authors showed that the PRX gene family is divided into six subfamilies according to the phylogenetical analysis of the conserved active site. Furthermore, they also confirmed that cis acting elements could effectively regulate the expression of PRX genes. Based on the results presented in this paper, most PRX members were situated in chloroplasts, and a few members were found in the cell membrane and nucleus.
This is an interesting study and the authors have collected a unique data-set using genomics and molecular genetics approaches. The paper is generally well written and structured. However, in my opinion the paper has some shortcomings in regards to some data analyses and text, and I feel this unique data-set has not been utilized to its full extent. In the text I have provided some remarks on the text. In several instances I also suggested to cite more relevant literature. Furthermore I made additional suggestions for more in-depth explanation of the analyses of the data. Given these shortcomings the manuscript requires minor revisions.

·

Basic reporting

This paper entitled "Genome-wide identification, evolution, regulation, and alternative shearing profile of peroxiredoxin genes in cotton" focused on the survey and characterization of PRX gene family in cotton species. This is an investigation study important to be performed in different cotton species, due to the low number of studies of the PRX family in plants. The manuscript is generally well written and provides new interesting information to cotton genome annotation. However, I have some suggestions for improvement:
(1) Section Material and Methods, at the final of the 'Plant material treatments and expression analysis' subsection I miss the description of experimental design. How was performed the experiment, completely randomized design or other? Please make this issue clear in the body of the text.
(2) Section Results, subsection 'Genome-wide identification and conserved motif analysis of the PRX gene family in cotton', line 223, change 'The MW and pI' to 'The Molecular Weight (MW) and Isoelectric point (pI)'
(3) Section Results, subsection 'Analysis of cis-acting elements in the promoter region', line 331, you speculate that PRX genes are sensitive to drought stress. However, there are many papers related to this issue. Please, rewrite the phrase or remove it.
(4) In the same section, what are the biological processes that PRX genes participate in?
(5) Supplementary files, please make available the figure S1 in better quality.

Experimental design

Please, see the general reporting

Validity of the findings

no comment

Additional comments

no comment

---

## Round 0.2 · Minor Revisions

The reviewers note that the manuscript is much improved but both have identified a few outstanding issues. Please revise your manuscript accordingly.

Reviewer 1 ·

Basic reporting

Compared to the initial version, Feng and collaborators have substantially improved the manuscript, notably regarding bibliography, writing style and classification of PRX genes. In the present version, they take in consideration the previous knowledge on these thiol-peroxidases. This is an important point since this modification emphasizes their original findings regarding the various isoforms in relation with Cys number and alternative splicing. Thus, their work provides valuable information about this class of peroxidases that fulfil key roles in maintenance of redox homeostasis in plants as well as in other organisms.

Experimental design

OK

Validity of the findings

Needs some improvement (see below)

Additional comments

Nonetheless, I have a few relatively important concerns that need to be addressed:

1) The writing style of the Abstract is still poor, since it has not been modified compared to the first version. It has to be revised in relation with the changes made in the manuscript.
- The first sentence is too simplistic. The authors should delete “in the body”.
- PRXs are now recognized not only to scavenge peroxides, but also to interact directly with other proteins and modify their redox status.
- Line 4: correct “families” to “family”
- The writing of the sentence “And differential...” needs to be improved.

2) Line 285-305: In this section, the authors already address subcellular localization, which is the topic of the last section in Results. They should move this part from the first section to the last one.

3) Regarding classification, the authors made substantial work to put it in relation with that performed by Prof. Dietz’s group. But there is still a rather surprising point. Several main PRXII types have been characterized in plants by many groups (PRXII-B, PRXII-E and PRXII-F). In cotton, the authors report about PRXII-B and PRXII-E isoforms, but nothing concerning PRXII-F, which is a well known mitochondrial isoform identified and characterized in various plant species (Arabidopsis, pea, tobacco, Acer..). Thus, it’s is very unlikely that there is no PRXII-F in cotton. Is this isoform classified within PRXII-B or PRX-IIE? Or is there another explanation? The authors have to answer and discuss this point. I checked myself sequences and found that some isoforms in the PRXII-E class (previously PRX5 type in the first manuscript) correspond to the PRXII-F type.

4) Regarding expression data, interpretation is not always convincing and the authors have to carefully check the whole part. For instance, the authors claim line 436 that expression of GhPRX14-D in root increases rapidly in response to salt (Fig. 7 N). This is not true, and nothing like the expression patterns of GhPRXs 11-D and 15-D, for which strong increases are indeed observed.

5) There are many errors in References and this is very confusing for the reader. For some, there are only initials (lines 52, 546, 574, 645, 775, 822). For others, first names and last names are reversed: Keisuke et al is in fact Yoshida et al (line 67), Mohamad-Javad et al corresponds to Vaseghi et al. (line 69), Pablo et al corresponds to Pulido et al (line 79), etc... The authors have to rigorously check all the references both in the text and in the list.

Other points:
- Line 57, Arabidopsis has not to be in Italics
- Lines 97-98: delete dots after Gossypium
- Lines 379, 380, 422 and throughout the manuscript: “”cis” in Italics.
- Line 277: replace “conservative” by “conserved”
- Line 389; replace drought-induced” by “drought-related”. Cis elements are not induced.
- Line 406: insert “one” after “only”
- -Line 416: what is the relationship between zein metabolism and cotton?
- Line 546: why “nitrate”?

·

Basic reporting

The paper entitled "Genome-wide identification, evolution, regulation, and alternative shearing profile of peroxiredoxin genes in cotton" focused on the survey and characterization of PRX gene family in cotton species, received a lot of improvement since from the last submission. I considered this paper ready for publication after some minor changes.
1. Please, remove the dots in the final of the genders in the lines 97 and 98.
2. In figure 7-8 is hard to see the different times due to small letters.

Experimental design

no comment

Validity of the findings

no comment

Additional comments

This study will be very helpful for a better annotation of cotton genomes and will help future studies on the PRX family.

---

## Round 0.3 · Minor Revisions

Please revise your manuscript according to the reviewers' comments.

Reviewer 1 ·

Basic reporting

In the title, I would suggest to replace "regulation" by "expression". This better reflects the content of the paper.
In Figure 2, correct at the top of the circle "PRXIIE" to "PRXIIF"
Abstract, lines 33-34. There is twice the same idea in the sentence that has to be revised Here is a suggestion for revision: Add "also" before "involved" and suppress ", but also interact directly... redox status".
Line 62: Replace "4 targeted chloroplasts" by "four were found to be targeted to chloroplasts".
Line 68: replace "Keisuke et al" by "Yoshida et al. (2018)".
Line 71: suppress "(Yoshida et al. 2018)"
Line 266: suppress "of the cotton PRX protein sequences".
Line 277: suppress "of the protein active sites".
Lines 284-285: revise by "The two catalytic Cys residues in PRXQ are separated..."
Line 290: replace "to" by "in".
Line 291: replace "which has..." by "and share the conserved sequence..."
Line 293: replace "which have...." by "and display a conserved motif (FG..)"
Line 332: corect to "Li"
Line 385: add "in terms of presence" after "second"
Line 403: correct to "MYB"
Line 460: what is the meaning of FPKM?
Line 464: suppress "only a"
Line 465: replace "...can cause...function," by " ... can modify protein sequence and function".
Line 504: suppress "Most".
Line 506: replace " and as a plastidial thiol-peroxidases" by "and are presumed to act as plastidial thiol oxidases".
Line 511: replace "PRXQ exists on" by "is localized on".
Line 564: replace "may be sensitive" by "respond".
Line 566: suppress "and MYB protein regulation". This is not comparable to stress or hormone treatments.
Line 569: replace "would like" by "aimed".
Line 573: suppress "Therefore"
Line 577: replace "the induction of" by "treatment with".
Lines 578-280: misspoken sentence. SA is not known to directly scavenge ROS, but to activate the expression of genes involved in this process. Please rephrase.
Lines 581-582: Awkward sentence. Do the authors mean that the genes belong to those having repressed expression?
Line 586: add "the expression of" before "10".
Line 591: replace "and drought will increase" by "which increases".
Line 597: replace "After" by "Since". delete "adversity".
Line 599: replace "identify the element" by "interact with these elements".
Line 606: replace "lowest" by "less".
Line 613: replace "as is the" by "such as".
Line 626: replace "cycle in" by "of".
Line 627: replace "signal transduction" by ", and transducing signal".
Line 627: suppress "and so on".
Line 637: add at the end of the sentence "and localization".
Lines 644-648; Repetition in the end of conclusions. Suppress: ". It aims to study... process of biological evolution".

Experimental design

No comment

Validity of the findings

No comment

Additional comments

Feng and collaborators have further improved the manuscript. The present version is globally satisfying. Nonetheless, the writing style and English language need to be carefully checked and some errors corrected.

·

Basic reporting

Dear Editor:
Title: Genome-wide identification, evolution, regulation, and alternative splicing profile of peroxiredoxin genes in cotton (#44008)

The manuscript by Feng et al, provides some very interesting results on the successful identification of PRX family genes in cotton. These results could be used to shed light on the functions of the genes of this family in this important plant. I can see that the authors have put a lot of effort to revise the manuscript. In general, I’m happy with the changes that the authors made to the paper and am suggesting that the paper is now acceptable for the publication.

Experimental design

The experimental apparatus is quite standard, and is appropriate for the study. There is no need to any do additional experiments to validate the results presented here as the results are very clear in the current format. I can also see that the authors also addressed my previous concerns on this section.

Validity of the findings

Again, I can see that the authors made changes to the discussion part as was suggested before and now I'm very happy with the validity of their results. The authors also discussed the implications of their experimental findings and answered my question on where they think research in this field should go from here.

Additional comments

The authors addressed all my concerns and implemented all suggestions in the current version of the manuscript. So, In general, I'm very satisfied now with the changes that the authors made to this manuscript and I highly recommend this paper for publication in Peer J Journal.

---

## Round 0.4 · Minor Revisions

The authors have taken the comments from reviewers into consideration, and made changes accordingly.

Before I can accept the manuscript, please attend to the following points raised by Gerard Laz, the Section Editor:

> The cotton genome is already well annotated and searching the database for "peroxiredoxin" returns a number of hits (see https://www.cottongen.org/). Are the genes identified in this manuscript the same as those already in the existing annotation? The authors need to acknowledge the current annotation, justify why a new search was needed, and discuss any differences between the set of genes that they identify as compared to the existing annotation.
>
> Line 456 claims that "most PRXs are differentially expressed" but no statistical analysis has be done. You cannot determine whether or not genes are differentially expressed from looking at a heat map without statistical analysis.
>
> Figure 9 and also abbreviations "alternative shearing"? Do the authors mean splicing?

---

## Round 0.5 · Minor Revisions

Your manuscript needs one quick fix. You use the term "alternative shearing" but I think you mean "alternative splicing". If you do mean alternative shearing, you should explain how this is different from alternative splicing and provide a reference."

---

## Round 0.6 · accepted · Accept

Your revised manuscript is acceptable.